# Reverberant 3D optical coherence elastography maps the elasticity of individual corneal layers

Fernando Zvietcovich[1], Pornthep Pongchalee[2], Panomsak Meemon[2], Jannick P. Rolland[3] & Kevin J. Parker[1]*

The elasticity mapping of individual layers in the cornea using non-destructive elastography techniques advances diagnosis and monitoring of ocular diseases and treatments in ophthalmology. However, transient Lamb waves, currently used in most dynamic optical coherence and ultrasound elastography techniques, diminish the translation of wave speed into shear/Young's modulus. Here, we present reverberant 3D optical coherence elastography (Rev3D-OCE), a novel approach leveraging the physical properties of diffuse fields in detecting elasticity gradients not only in the lateral direction, but also along the depth axis of the cornea. A Monte Carlo analysis, finite element simulations, and experiments in layered phantoms are conducted to validate the technique and to characterize the axial elastography resolution. Experiments in ex vivo porcine cornea at different intraocular pressures reveal that Rev3D-OCE enables the elastic characterization of single layers that matches the anatomical description of corneal layers with unprecedented contrast in the dynamic OCE field.

[1] Department of Electrical and Computer Engineering, University of Rochester, Rochester, NY 14627-0126, USA. [2] Suranaree University of Technology, Nakhon Ratchasima, Thailand 30000. [3] The Institute of Optics, University of Rochester, Rochester, NY 14627-0186, USA. *email: kevin.parker@rochester.edu

The cornea is one of the most critical parts of the eye since it is responsible for almost two-thirds of the eye's total refracting power[1]. The topography of corneal layers and surfaces has a direct impact on corneal optical aberrations such as defocus, astigmatism, and spherical aberrations, which can significantly diminish the optical performance of the eye[2]. Sensitivity studies demonstrate that the stiffness of the cornea, among other factors, is the most significant biomechanical parameter affecting corneal optical behavior[3]. Consequently, alterations of mechanical properties of the cornea (such as elasticity and viscosity) produced by ocular diseases, interventions, or treatments have a direct impact on the vision quality.

The cornea is a highly organized tissue containing at least five layers with differentiated structure, mechanical properties, and physiological functions: epithelium, Bowman's membrane, stroma, Descemet's membrane, and endothelium[4]. Various corneal dystrophies, such as keratoconus, may affect single layers[5–7] or all the layers of the cornea[8]. Current treatments such as LASIK refractive surgery, UV-A/riboflavin collagen crosslinking, and laser therapy can modify the mechanical properties of individual or all layers[9–11]. It has also been demonstrated that intraocular pressure (IOP) has an important effect on the elasticity measurements of the cornea due to the non-linear properties of this tissue type[12,13]. This IOP-elasticity relationship in cornea can be a critical component in the early diagnosis and monitoring of open-angle glaucoma[14]. In addition, there is an increasing interest in computational models to simulate patient-specific corneal behavior based on accurate mechanical characterization with useful applications in the prediction of LASIK outcomes[15,16] and clinical decision-making[4,17,18]. Therefore, the elastic characterization of individual corneal layers using non-destructive techniques is of great importance in ophthalmology, not only for advancing the understanding, diagnosis, and monitoring of ocular diseases and treatments, but also for the successful modeling of patient-specific corneas, and the development of synthetic and tissue-engineered corneal substitutes[19,20].

The emergence of elastography as a nondestructive method for the measurement of biomechanical properties of tissue began with the development of ultrasound elastography (USE)[21]. One of the earliest applications of USE in corneal layers was implemented by Hollman et al. who calculated depth-dependent strain profiles of the cornea using high frequency ultrasound imaging[22]. Other USE approaches used acoustic radiation force to produce and measure displacement[23] and shear wave propagation[24–26] in the cornea. Due to limitations in spatial resolution, displacement sensitivity, and the need for a coupling material such as gel or water, the application of USE in the cornea has been very restrictive.

Optical coherence tomography (OCT) was later developed as an imaging technique capable of obtaining images with micrometer resolution (~2 to 10 μm) at high acquisition speeds (~20 kHz to 1 MHz A-line rate) in a minimally invasive fashion (no need for coupling material)[27]. Elastography was implemented in OCT for the first time by Schmitt et al. and this modality was termed optical coherence elastography (OCE)[28]. One of the earliest applications of OCE in cornea was conducted by Nahas et al., in which corneal strain maps were extracted by compressing the tissue using a piston and detecting corneal displacement using a full-field OCT system[29]. Although this technique provides an important contrast in the detection of corneal layers, the absence of force measurements prevents the calculation of qualitative biomechanical parameters such as elasticity in the form of Young's or shear moduli[30]. Finally, Brillouin scattering-based optical imaging technique provides high contrast detection of layers in cornea (i.e., at least

four corneal layers are detected)[31–33]. However, obtaining quantitative biomechanical parameters such as shear or Young's modulus from Brillouin shifts is still an unsolved problem[34].

Later, the implementation of shear wave-based techniques in OCT[35–37] enabled the study of wave propagation with an unprecedented level of detail in corneal studies. Wang et al. then presented a dynamic OCE method for the characterization of corneal layers based on Lamb wave propagation (also called the Lamb wave method in this paper)[38]. The Lamb wave method has since been implemented in diverse studies including cornea anisotropy[39], the impact of therapy on corneal elasticity[40,41], and the relationship between Lamb wave speed and IOP levels[42]. However, a Lamb wave is a type of surface acoustic wave that propagates in thin shells[43]. It is speed-dispersive, and requires knowledge of the shell thickness, boundary conditions, excitation frequency, and direction of wave propagation in order to apply inverse approaches to convert Lamb wave speed into shear/Young's modulus[43,44]. This application becomes more complex when considering the multiple corneal layers and the viscoelastic properties of porcine and human corneal tissue[45,46]. Therefore, due to the complexity of the corneal boundary conditions, depth-dependent information of elasticity in the cornea cannot yet be provided.

In wave-based OCE, reverberant or diffuse fields[47,48] are produced by the interaction of multiple shear waves traveling in a statistical distribution across all possible random directions. These fields promote the propagation of a higher ratio of shear waves in tissue compared to a transient excitation in which surface acoustic waves (i.e., Lamb waves for the cornea case) are the predominant perturbation[49,50]. The propagation of shear waves in the cornea produced by reverberant fields is desired since shear wave speed can be correlated to local shear modulus in a straightforward fashion, avoiding the undesirable effect of Lamb waves.

In this study, we present reverberant 3D optical coherence elastography (Rev3D-OCE), leveraging the physical properties of diffuse fields in detecting elasticity gradients along the depth axis of the cornea. A Monte Carlo analysis, finite element simulation, and experiments in layered phantoms are conducted to validate the key theoretical principles. Results demonstrate the detection of differentiated elastic layers in a 3D model of the cornea, and quantify the axial elastography resolution. Additional experiments in ex vivo porcine cornea at different IOP levels reveal that Rev3D-OCE enabled the elastic characterization of each single layer with unprecedented contrast in the dynamic OCE field that matches the anatomical description of the corneal layers. Results are compared with the conventional Lamb wave method.

## Results

**Principles of Rev3D-OCE and Monte Carlo analysis**. A Monte Carlo analysis of a reverberant shear wave field applied to a uniform linear-elastic medium validates our proposed theoretical model and equations, fundamental for the local estimation of shear wave speed. A reverberant field can be understood as the superposition of plane shear waves traveling in random directions[48]. Shear waves are transversal body-type waves; therefore, the particle velocity produced by these perturbations is perpendicular to the direction of propagation. For a given 3D point $\boldsymbol{\varepsilon}$ in the Cartesian system, three orthogonal vectors in the spherical coordinate system are found: $\hat{\mathbf{r}}$, $\hat{\boldsymbol{\theta}}$, and $\hat{\boldsymbol{\varphi}}$ as shown in Fig. 1a. Then, the corresponding particle velocity field $\mathbf{V}(\boldsymbol{\varepsilon}, t)$, at position $\boldsymbol{\varepsilon}$ and time $t$, in a reverberant chamber produced by plane waves propagating with a wave number $k$ and radial frequency $\omega_0$ is

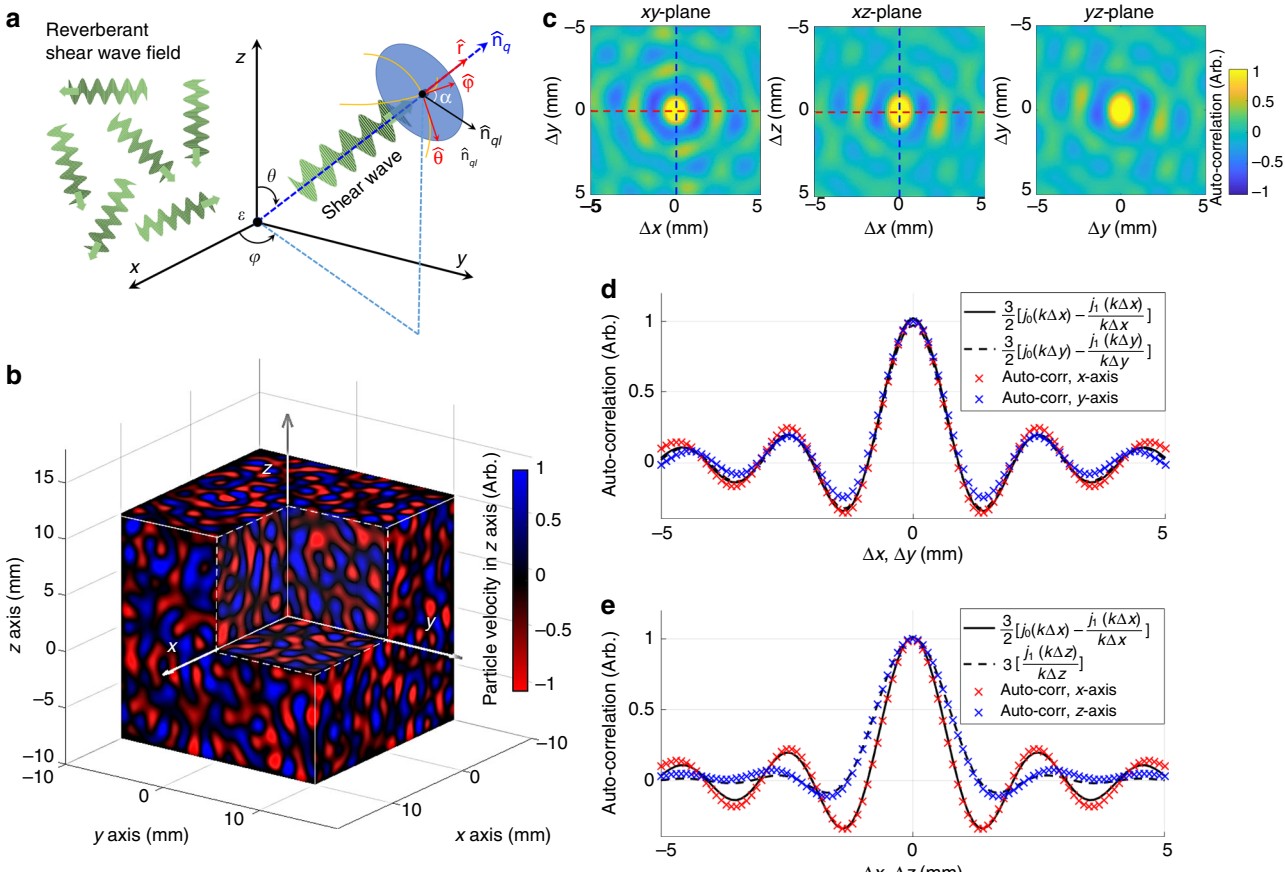

**Fig. 1** Principles of reverberant fields in elastic media. Underlying principle of Rev3D-OCE and theoretical predictions of the auto-correlation function in a reverberant field. **a** Graphic illustration of a shear wave reverberant field modeled as a superposition of plane shear waves traveling in a random direction $\hat{\mathbf{n}}_q$, with perpendicular particle velocity $\hat{\mathbf{n}}_{ql}$ parallel to the plane (blue disk) formed by basis vectors $\hat{\boldsymbol{\theta}}$ and $\hat{\boldsymbol{\varphi}}$, and a particle velocity random amplitude $v_{ql}$. **b** Particle velocity field along z-axis $V_z(\boldsymbol{\varepsilon}, t_0)$ at instant $t_0$ in a cube of $20 \times 20 \times 20$ mm³ of uniform elastic material after conducting a Monte Carlo analysis of reverberant shear waves with frequency of $f_0 = 2$ kHz, and a constant shear wave speed $c_s = 4$ m s⁻¹ after a total of 10,000 realizations of all random variables. **c** Real part of 2D auto-correlation maps extracted from the reverberant volume $V_z(\boldsymbol{\varepsilon}, t_0)$ in (**b**) taken in three planes: xy-plane, xz-plane, and yz-plane. Colorbar represents normalized auto-correlation in arbitrary units. In (**d**) and (**e**), auto-correlation profile cuts in both axis extracted from (**c**) in the xy-plane and xz-plane, respectively, are plotted with analytical Eqs. (2) and (3) for further comparison. Auto-correlation profiles plots in yz-plane are similar to xz-plane since both x and y axes are perpendicular to measurement axis $\hat{\mathbf{e}}_z$

modeled as

$$\mathbf{V}(\boldsymbol{\varepsilon}, t) = \sum_{q,l} \hat{\mathbf{n}}_{ql} v_{ql} e^{i(k\hat{\mathbf{n}}_q \cdot \boldsymbol{\varepsilon} - \omega_0 t)} \quad (1)$$

where the index $q$ represents a realization of the random unit vector $\hat{\mathbf{n}}_q$ describing the direction of wave propagation, and the index $l$ represents a realization of the random unit vector $\hat{\mathbf{n}}_{ql}$ describing direction of particle velocity parallel to the disk formed by the basis vectors $\hat{\boldsymbol{\theta}}$ and $\hat{\boldsymbol{\varphi}}$ defined within a realization of $q$ (see Fig. 1a). The particle velocities $v_{ql}$ are an independent, identically distributed random variable describing the magnitude of particle velocity within a realization of $q$ The summation of $q$ is understood to be taken over the $4\pi$ solid angle, and the summation of $l$ is taken over a $2\pi$ angle within the disk.

If the particle velocity in Eq. (1) is measured along the z-axis, then $V_z(\boldsymbol{\varepsilon}, t) = \mathbf{V}(\boldsymbol{\varepsilon}, t) \cdot \hat{\mathbf{e}}_z$, where $\hat{\mathbf{e}}_z$ is a unit vector in the z direction. Figure 1b shows $V_z(\boldsymbol{\varepsilon}, t_0)$ for an instant $t_0$ in a cube of $20 \times 20 \times 20$ mm³ of uniform elastic material after conducting a Monte Carlo analysis. Four uniformly distributed random variables were considered: angles $\varphi$ and $\theta$, each one covering a range of $[0, 2\pi]$ radians, for the definition of $\hat{\mathbf{n}}_q$; angle $\alpha$, covering a range of $[0, 2\pi]$ radians for the definition of $\hat{\mathbf{n}}_{ql}$; and the scalar value $v_{ql}$, covering a range of $[-v_{max}, v_{max}]$, where $v_{max}$ is the

maximum particle velocity amplitude supported in the system. In the analysis, the frequency of shear waves was defined as $f_0 = 2$ kHz, leading to $\omega_0 = 2\pi f_0$, and a constant shear wave speed of $c_s = 4$ ms⁻¹ within the cube, where $c_s = \omega_0/k$. A total of 10,000 realizations of all random variables were considered in order to recreate the reverberant field. Detailed description about the Monte Carlos Analysis can be found in the Methods section.

Given the reverberant field $V_z(\boldsymbol{\varepsilon}, t_0)$ shown in Fig. 1b, closed-form complex analytical solutions can be found for the spatial auto-correlation of such fields along parallel and orthogonal directions to the measurement axis $\hat{\mathbf{e}}_z$. In the orthogonal case, the auto-correlation of the reverberant field along the x-axis (or y-axis) is given by (complete theoretical derivations are explained in Supplementary Methods):

$$B_{V_z V_z}(\Delta \varepsilon_x) = \frac{\beta}{2} \left[ j_0(k\Delta \varepsilon_x) - \frac{j_1(k\Delta \varepsilon_x)}{k\Delta \varepsilon_x} \right] \quad (2)$$

where $\beta$ is the expected value of squared particle velocity magnitude $v_{ql}^2$ over both $q$ and $l$ realizations; $j_0$ and $j_1$ are spherical Bessel functions of the first kind of zero and first order, respectively; and $\Delta \varepsilon_x = \Delta \boldsymbol{\varepsilon} \cdot \hat{\mathbf{e}}_x$, with $\hat{\mathbf{e}}_x$ as a unit vector along the x-axis. It is important to note that $B_{V_z V_z}(\Delta \varepsilon_x)$ and $B_{V_z V_z}(\Delta \varepsilon_y)$ have the same solution since both correlation axis $\hat{\mathbf{e}}_x$, and $\hat{\mathbf{e}}_y$ are

perpendicular to measurement axis $\hat{\mathbf{e}}_z$. Therefore, $B_{V_z V_z}(\Delta\boldsymbol{\varepsilon})$ is symmetric around the origin for any $\Delta\boldsymbol{\varepsilon}$ such that $\Delta\boldsymbol{\varepsilon} \cdot \hat{\mathbf{e}}_z = 0$.

Finally, the parallel case, the spatial auto-correlation of the reverberant field is taken along the z-axis and is given by (complete theoretical derivation are explained in Supplementary Methods):

$$B_{V_z V_z}(\Delta\varepsilon_z) = \frac{\beta}{2}\left[\frac{j_1(k\Delta\varepsilon_z)}{k\Delta\varepsilon_z}\right] \tag{3}$$

Figure 1c shows the spatial 2D auto-correlation of the reverberant volume $V_z(\boldsymbol{\varepsilon}, t_0)$ taken in three planes: (1) xy-plane, (2) xz-plane, and (3) yz-plane. As expected, case 1 is axisymmetric since both the x and y axis are perpendicular to the measurement axis $\hat{\mathbf{e}}_z$. Profile cuts in both axes are plotted in Fig. 1d, and follow Eq. (2), which confirms the accuracy of the model compared against the Monte Carlo analysis. For cases 2 and 3, the symmetry is broken since the z-axis is parallel to the measurement axis $\hat{\mathbf{e}}_z$ and its profile cut (Fig. 1e) follows Eq. (3), while the x or y axis are perpendicular to $\hat{\mathbf{e}}_z$, giving the same solution as in Eq. (2). The local wave number $k^*$ can be estimated by taking the 2D auto-correlation in a region of a reverberant field and fitting auto-correlation profiles to Eqs. (2) and (3) according to the plane case. Therefore, given the excitation frequency $\omega_0$, shear wave speed is calculated using $c_s = \omega_0/k^*$; and, assuming a homogeneous and isotropic material, shear modulus can be calculated as[30]

$$G = c_s^2 \rho \tag{4}$$

where $G$ is the shear modulus, and $\rho$ is the density of the material. Furthermore, if a transverse isotropic model of the material is assumed, $G$ corresponds to out-of-plane shear modulus (shear modulus parallel to the corneal depth), according to the terminology provided by Ramirez-Garcia et al.[17].

**Validation of Rev3D-OCE using finite element analysis.** Numerical simulations using a finite element model of the cornea with four differentiated elastic layers (A, B, C, and D) demonstrates the capabilities of the Rev3D-OCE method in layer detection compared to the conventional Lamb wave method. The layers were monotonically distributed from stiffer (A) to softer (D) using Young's moduli values ("Methods" section) based on the decreasing depth-dependent wave speed tendency reported in Wang et al.[38] for porcine cornea. The diagram in Fig. 2a-left, shows a 3D volume of cornea subjected to sinusoidal steady-state excitation of 2 kHz through eight circular-distributed force heads. As expected, wavelength changes in particle velocity along depth (z-axis) are detected and transformed into shear wave speed values using the reverberant theory (Eqs. (1)–(3)), and shown in Fig. 2a-right. In addition, shear wave speed values for each layer (A, B, C, D) are in good agreement with the layer material properties (2.5, 2, 1.3, 0.8 m s$^{-1}$), respectively, set in the simulation ("Methods" section).

The Lamb wave method was also simulated by applying an axisymmetric Gaussian ($\sigma = 0.3$ mm) transient ($t = 1$ ms) force in the center of the layered cornea model producing a cylindrically-shaped Lamb wave propagation as shown in Fig. 2b-left. Fourier analysis applied to space-time representations of wave propagation at each layer indicates that the antisymmetric zero order Lamb wave is the most predominant given the excitation shape and boundary conditions. Fig. 2b-right shows dispersion curves of Lamb wave speed calculated in each layer. In addition, we included dispersion curves when a transient $t = 0.5$ ms force (same Gaussian extend of $\sigma = 0.3$ mm) is applied to the same model in order to explore the effects of higher temporal frequencies. The Fourier transform of both temporal excitations

($t = 1$ ms and $t = 0.5$ ms) measured in the cornea model is shown in Fig. 2b-right. Results confirmed the antisymmetric zero order behavior during Lamb wave propagation. As expected, for a selected frequency, the depth-dependent variation of Lamb wave speed is small compared to the Rev3D-OCE method shown in Fig. 2a-right. In addition, the Lamb wave speed depth profile has a decreasing tendency due to the high-low layer elasticity configuration. However, speed values are not in complete agreement with the layer material properties set in the simulation.

**Axial elastography resolution characterization of Rev3D-OCE.** Experimental results obtained in horizontally-layered gelatin phantoms show that Rev3D-OCE has an accuracy error of <7% and an axial elastography resolution of $\delta_z = 55.5 \mu m \pm 17.3 \mu m$, measured along the average transition from the softer top (3% gelatin concentration) to stiffer bottom (5% gelatin concentration) layers distributed as shown in Fig. 3a-left. The average thickness of the top layer was measured to be ~0.3 mm and the bottom layer is considered as semi-infinite media. The phantom was subjected to a reverberant shear wave field of 2 kHz using a piezoelectric actuator attached to a 3D-printed ring with eight heads slightly touching the phantom surface and the motion (particle velocity field shown in Fig. 3a-left) was detected using a spectral-domain phase sensitive optical coherence tomography (PhS-OCT) system (details of the system setup are explained in the Methods Section). Figure 3b shows two motion frames extracted at the top 3% layer (depth $z_0 = 0.21$ mm) and bottom 5% layer (depth $z_0 = 0.69$ mm) with their respective 2D auto-correlation plots $B_{Vz Vz}(\Delta\varepsilon_x, \Delta\varepsilon_y)$ calculated within a $0.8 \times 0.8$ mm$^2$ window and indicating differentiated material properties.

Layer differentiation is confirmed when the average auto-correlation curve, taken along radial cuts of the 2D auto-correlation plot covering 360 degrees, was fitted to the theoretical model (Eq. (2)) of reverberant fields in elastic media for the estimation of global wavenumbers $k_{3\%} = 10.68$ krad m$^{-1}$ and $k_{5\%} = 6.63$ for the 3% and 5% motion frames, respectively, as shown in Fig. 3c. Auto-correlation measurements ($N = 5$) were taken at different regions of the layered phantom, resulting in an average shear wave speed of $c_{3\%} = 1.17 \pm 0.14$ m s$^{-1}$ and $c_{5\%} = 1.89 \pm 0.31$ m s$^{-1}$ in the 3% and 5% phantom material layer, respectively, using $c_s = \omega_0/k$. Experimental results are consistent with material ground truth parameters $c_{3\%}^{MM} = 1.15 \pm 0.12$ m s$^{-1}$ (1.74% accuracy error) and $c_{5\%}^{MM} = 2.03 \pm 0.14$ m s$^{-1}$ (6.89% accuracy error) obtained using mechanical measurements (MM) of samples of the same gelatin concentration (details of phantom characterization in "Methods" section). Finally, the average depth-dependent shear wave speed profile describing the transition from the top (3%) to the bottom (5%) layers (Fig. 3d) was obtained by analyzing every motion frame along depth using the same methodology described previously. The speed profile was fitted to a sigmoid function described as

$$c(z) = c_{5\%} + (c_{3\%} - c_{5\%})\frac{1}{1 + e^{(z-z_0)/\tau}} \tag{5}$$

where $c_{3\%}$ and $c_{5\%}$ are the average shear wave speeds in the 3% and 5% media, respectively; $z_0$ and $\tau$ represent the location and width of the transition, respectively. Then, we calculated the average full-width half-maximum (FWHM) of the derivate of $c(z)$ with respect to $z$, from the 3 to 5% layers: $55.5 \mu m \pm 17.3 \mu m$ as shown in Fig. 3d.

**Layer mapping of ex vivo porcine cornea using Rev3D-OCE.** The Rev3D-OCE method enabled the elastic characterization of single layers in ex vivo porcine cornea with unprecedented contrast in the dynamic OCE field that matches the anatomical

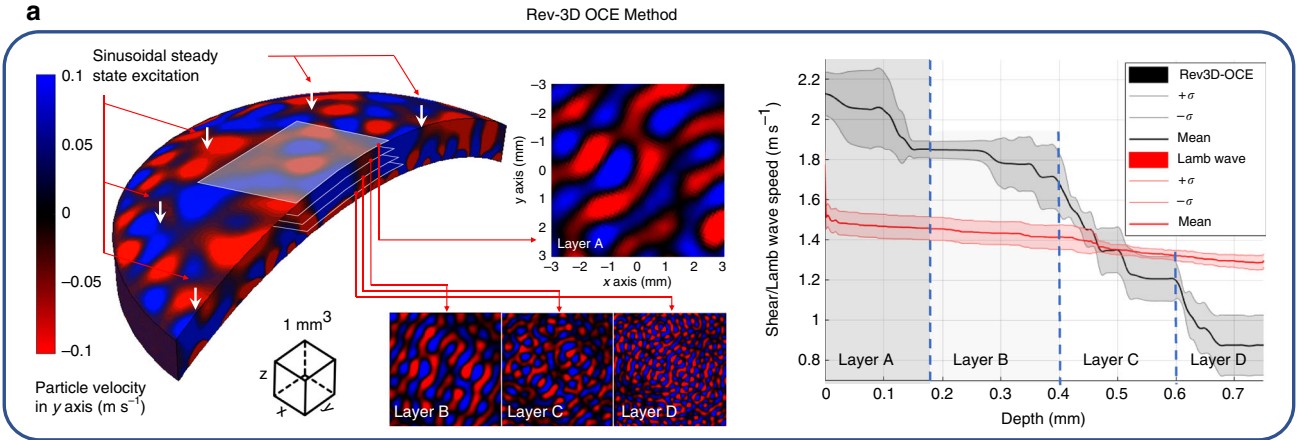

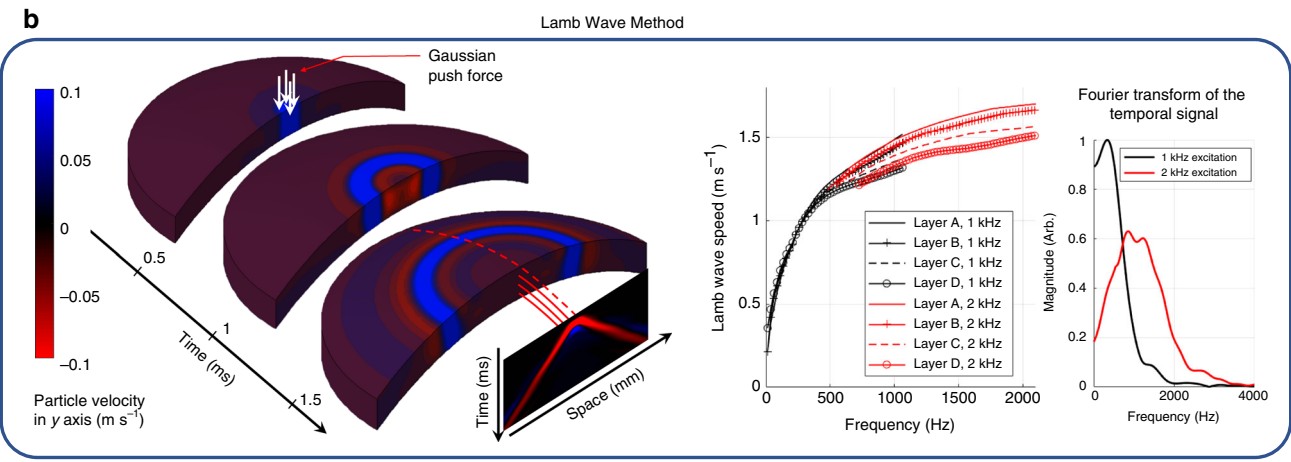

**Fig. 2** Comparative analysis of Rev3D-OCE and the Lamb wave method using numerical simulations. Numerical simulation results of a 3D finite element model of cornea composed of four layers with differentiated linear elastic materials distributed monotonically from stiffer to softer: A, B, C, and D, corresponding to 2.5, 2, 1.3, and 0.8 m s$^{-1}$ of shear wave speed. **a** Cornea model subjected to sinusoidal steady-state excitation of 2 kHz (Rev3D-OCE method) through eight circular-distributed force heads. Reverberant patterns extracted in each layer depict differentiated wavelength and are transformed into shear wave speed values using Eqs. (1)–(3) as plotted in the shear/Lamb speed vs. depth diagram. **b** Cornea model subjected to axisymmetric Gaussian ($\sigma = 0.3$ mm) transient ($t = 1$ ms) force (Lamb wave method) producing a cylindrically-shaped Lamb wave propagation. Dispersion curves (Lamb speed vs. frequency) are calculated for each layer. Dispersion curves produced from a transient ($t = 0.5$ ms, $\sigma = 0.3$ mm) force are also included. The Fourier transform of both temporal excitations ($t = 0.5$ ms, and $t = 1$ ms) measured in the model are also shown. Given a selected frequency (1 kHz), depth-dependent Lamb wave speed profile is plotted in the shear/Lamb speed vs. depth diagram of (**a**) for further comparison

description of corneal layers. A fresh porcine eye cornea was subjected to a reverberant shear wave field of 2 kHz using a piezoelectric actuator attached to a 3D-printed ring with eight heads slightly touching the cornea surface and the motion (particle velocity) was detected using a spectral-domain PhS-OCT system (details of the system setup are explained in the "Methods" section). After the acquisition, a 3D reverberant cornea volume was obtained and shown in Fig. 4a. Reverberant frames of particle velocity are extracted following the surface of the cornea at all depths and the corneal curvature was compensated for. Figure 4b shows a reverberant frame extracted at the top surface (epithelium), the 2D auto-correlation plot calculated within a 2 × 2 mm$^2$ window in the motion frame, and the average auto-correlation curve ($N = 360$ curves) taken along radial cuts of the 2D auto-correlation plot covering 360 degrees, with shaded regions indicating standard error (SE). The theoretical reverberant model (Eq. 2) was fitted to the average profile for the estimation of local wavenumber ($k^*$), and, subsequently, shear wave speed ($c_s$) using $c_s = \omega_0/k^*$.

In the context of corneal layer characterization, we de-emphasized lateral elastography resolution by averaging all speed values calculated within a single reverberant frame using a 2 × 2 mm$^2$ window, making sure the window captures at least half wavelength of the reverberant field pattern in all cases. We also wanted to characterize SE as a measurement of anisotropy within each corneal layer. Therefore, shear speed values (mean ± SE) were calculated for all frames from the top (epithelium) to the bottom (endothelium) layers of the cornea (detailed signal processing scheme description in "Methods" section). Figure 4c shows the average depth-dependent shear wave speed profile of cornea and how it correlates with (1) structural information from the B-mode intensity image of the same cornea sample, and (2) the anatomical description of some of the corneal layers. Shear wave speed values for a selected cornea span from 2 m/s to 4 m/s (2 m/s min-to-max range) which is equivalent to 4 kPa and 16 kPa of shear modulus, respectively, if a density of $\rho = 1000$ kg m$^{-3}$ is assumed in Eq. (4). In Fig. 4d, the depth-dependent shear wave speed was put into a 2D format and superimposed on the B-mode structural image of the cornea. The level of contrast in layer detection of cornea achieved with the Rev3D-OCE method is unprecedented in the dynamic wave-based OCE field. Previous work using transient Lamb waves[38,41]

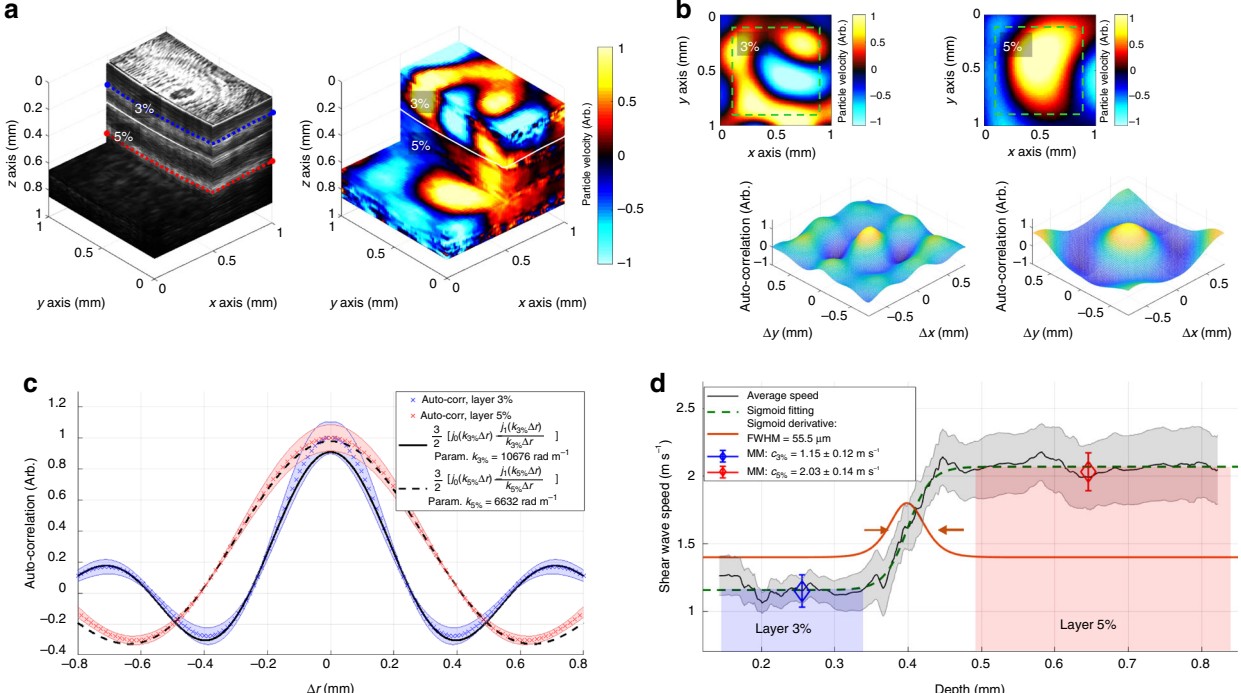

**Fig. 3** Axial elastography resolution characterization of Rev3D-OCE. Experimental results of Rev3D-OCE in layered phantoms for axial elastography resolution characterization. **a** In the left, 3D structural $1 \times 1 \times 1\,mm^3$ volume of a horizontally-distributed layer phantom: softer top layer (3% gelatin concentration, ~0.3 mm average tickness), and stiffer bottom layer (5% gelatin concentration, considered semi-infinite). Colormap in gray scale represents intensity in arbitrary units. In the right, a reverberant motion volume (particle velocity in arbitrary units) of the same phantom is shown. **b** Motion frames extracted from motion volume in (**a**) at the 3% region ($z_0 = 0.21\,mm$, blue dashed cut in (**a**)-left), and 5% region ($z_0 = 0.69\,mm$, red dashed cut in (**a**)-left). 2D auto-correlation plots are calculated within a $0.8 \times 0.8\,mm^2$ window of each motion profile. **c** Average of auto-correlation curves ($N = 360$ curves) taken along radial cuts of each 2D auto-correlation plot in (**b**) covering 360 degrees, with shaded regions indicating standard error (SE). Equation (2) is fitted to each average auto-correlation curve demonstrating a differentiated elasticity between the 3% ($k_{3\%} = 10.68\,krad\,m^{-1}$) and 5% ($k_{5\%} = 6.63$ $krad\,m^{-1}$) layers. **d** Average depth-dependent shear wave speed profile ($N = 5$ independent measurements at different regions of the layered phantom) describing the speed transition from the top 3% to the bottom 5% layers. Mean ± SE. The speed profile is fitted to a sigmoid function (Eq. 5) for the calculation of the average full-width half-maximum layer transition: 55.5 μm ± 17.3 μm. Ground truth speed parameters ($c_{3\%}^{MM} = 1.15 \pm 0.12\,m\,s^{-1}$ and $c_{5\%}^{MM} = 2.03 \pm 0.14\,m\,s^{-1}$) obtained using mechanical testing ($N = 3$ samples per gelatin concentration) show good agreement with the speed profile in the uniform regions

shows depth-dependent wave speed profiles with reduced contrast (<0.3 m/s min-to-max range), and fewer numbers of features consistent with the anatomical description of corneal layers. Nevertheless, its decreasing tendency is in agreement with results reported by Rev3D-OCE and other approaches[22,23,29,31] (further comments in "Discussion").

**Rev3D-OCE and Lamb wave methods vs. IOP of cornea**. Experiments in an IOP-controlled porcine eye cornea using the Rev3D-OCE and the conventional Lamb wave method for the detection of elastic layers demonstrate that Rev3D-OCE is able to monitor single corneal layer changes for different IOP levels. In Fig. 5a, we show reverberant volumes of cornea subjected to four IOP levels: 10.24, 15.80, 20.50, and 25.25 mmHg. The increasing tendency of wavelength in the reverberant patterns when the IOP level increases confirms the non-linear elastic nature of corneal tissue. Reverberant frames of particle velocity were extracted, following the curved shape of the cornea at all depths, and processed using auto-correlation ($2 \times 2\,mm^2$ window) and the theoretical model (Eq. (2)) of reverberant fields in elastic media. Then, local shear wave speed ($c_s$) was calculated for all depths from the top (epithelium) to the bottom (endothelium) layers of the cornea. Figure 5b shows shear wave speed volumes for each IOP level. It is evident that the shear wave speed increases with IOP. The min-to-max range of speed change due to the application of different IOP levels was reported to be 3.5 m s$^{-1}$.

Depth-dependent shear wave speed profiles of cornea subjected to different IOP levels were calculated by averaging all speed values within every single layer in all volumes of Fig. 5b in order to increase accuracy in layer characterization. Mean ± SE measurements for each layer are reported in Fig. 6a. In order to explore the relationship between Rev3D-OCE and the conventional Lamb wave method, depth-dependent group speed profiles were obtained by propagating Lamb waves in the same cornea samples. Fourier analysis was applied to space-time representations of Lamb wave propagation at all layers for each IOP case, as shown in Fig. 6b. The shape of the dispersion curves indicates that the antisymmetric zero-order Lamb wave is the most predominant given the excitation shape and boundary conditions. This finding is in complete agreement with the simulations using a finite element analysis reported in Results Section. Cut-off frequencies of dispersion curves in Fig. 6b are based on the signal-to-noise ratio of frequency samples. Due to the nature of the Rev3D-OCE method, a single excitation frequency should be selected for further comparison with the Lamb wave method. Then, a frequency $f_0 = 800\,Hz$ or $\omega_0 = 1600\,\pi\,rad\,s^{-1}$ was chosen in order to cover for all four IOP cases in Fig. 6b, and depth-dependent Lamb wave speed profiles were plotted in Fig. 6a together with the average shear wave speed profiles of the same

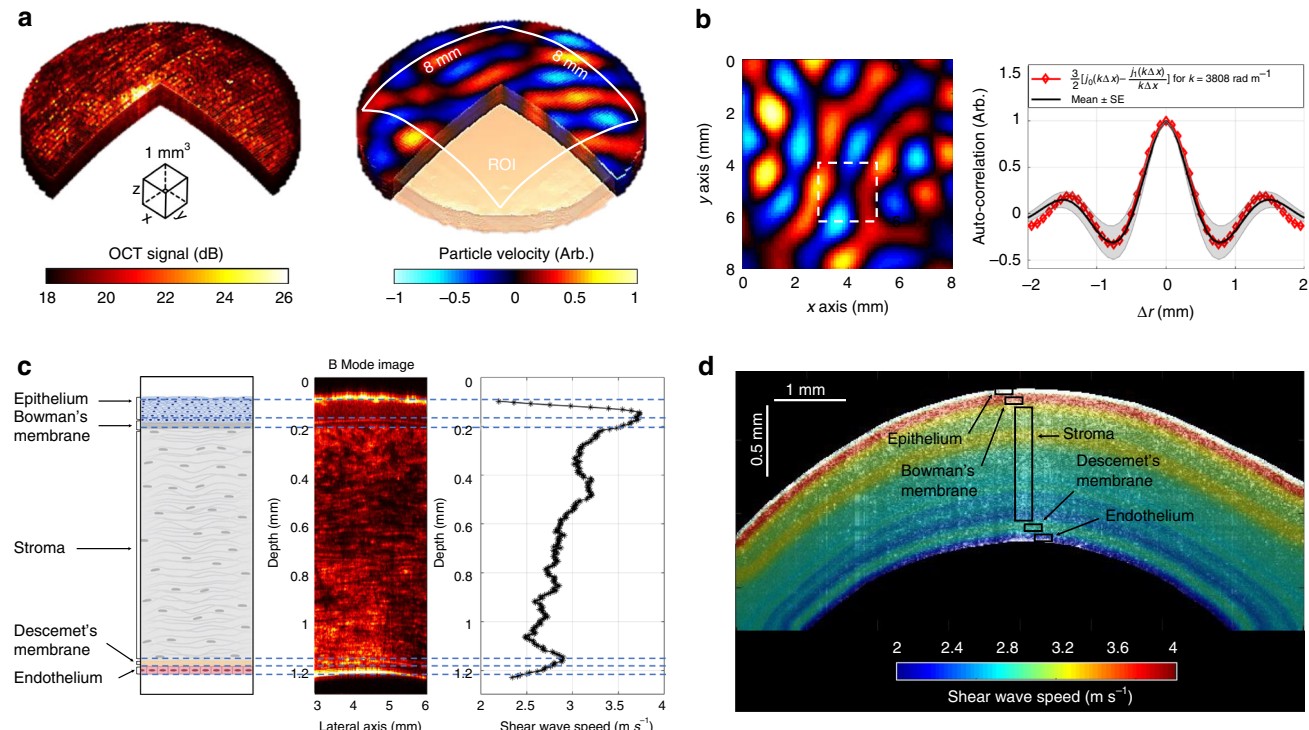

**Fig. 4** Elasticity characterization of layers in ex vivo porcine cornea using Rev3D-OCE. Rev3D-OCE results in ex vivo porcine cornea using a 2 kHz sinusoidal steady-state excitation and a spectral domain PhS-OCT system. **a** Acquired B-mode 3D volume, and particle velocity reverberant volume of cornea are shown in left and right side, respectively. In the left side of (**b**), a particle velocity frame is extracted at the cornea epithelium in (**a**) and compensated for the cornea curvature. Colormap represents particle velocity in arbitrary units as indicated in the colorbar in (**a**). The average auto-correlation curve ($N = 360$ curves) taken along radial cuts of the 2D auto-correlation plot ($2 \times 2\,mm^2$ window size) covering 360 degrees was fitted to Eq. (2) and shown in the right side. Mean ± SE. Then, estimations of local wavenumber ($k^*$), and, therefore, shear wave speed ($c_s$) using $c_s = \omega_0/k^*$, were calculated for all depths from the top (epithelium) to bottom (endothelium) layers of cornea. **c** Average depth-dependent shear wave speed profile of cornea and how it correlates with (1) structural information from the B-mode intensity image (colorbar as in (**a**)) of the same cornea sample, and (2) the anatomical description of some of the corneal layers. **d** 2D shear wave speed map superimposed on a B-mode structural image of the cornea

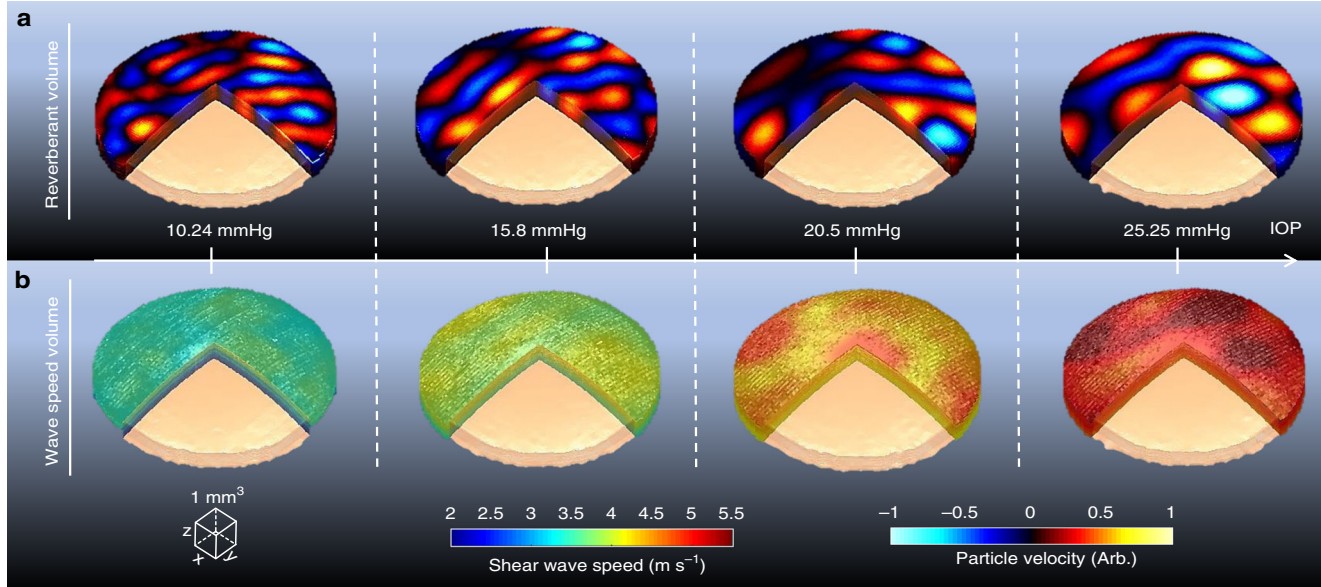

**Fig. 5** Volumetric analysis of ex vivo IOP-controlled porcine corneas using Rev3D-OCE. **a** Particle velocity reverberant volumes of cornea subjected to four IOP levels: 10.24, 15.80, 20.50, and 25.25 mmHg. Particle velocity frames are extracted, following the curved shape of the cornea at all depths, and processed using auto-correlation ($2 \times 2\,mm^2$ window size) and fitted to Eq. (2) for the estimation of local wavenumber ($k^*$). b Shear wave speed volumes of the cornea for each IOP case calculated using $c_s = \omega_0/k^*$

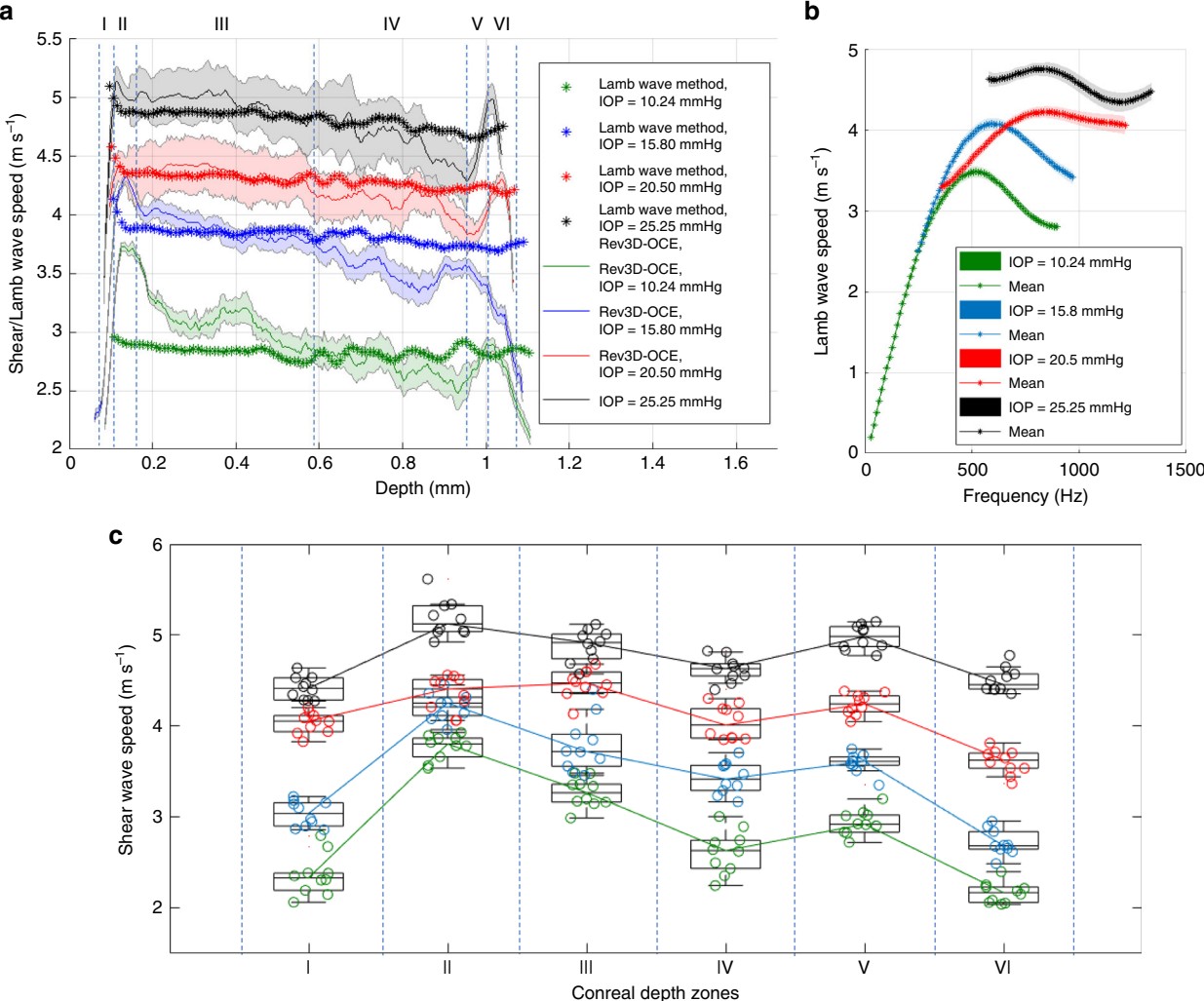

**Fig. 6** Statistical analysis of depth-dependent wave speed profiles of ex vivo IOP-controlled porcine corneas. **a** Comparison of average depth-dependent wave speed profiles using Rev3D-OCE and the Lamb wave method in a single cornea subjected to four IOP levels: 10.24, 15.80, 20.50, and 25.25 mmHg. Mean ± SE within every single layer. Lamb wave speed profiles were calculated at a frequency $f_0 = 800$ Hz to cover for all four IOP cases in (**b**). **b** Dispersion analysis of wave speed using the Lamb wave method in a cornea subjected to four IOP levels. Curves indicate that the antisymmetric zero-order Lamb wave is the most predominant given the excitation shape and boundary conditions. **c** Repeatability study ($N = 10$ corneas) of depth-dependent shear wave speed profiles divided into six zones: I (epithelium), II (Bowman's membrane), III (upper stroma), IV (lower stroma), V (Descemet's membrane), and VI (endothelium), subjected to four IOP levels. Horizontal lines within boxes indicate median values, boxes denote interquartile ranges. Whiskers on the box plots span one standard deviation

cornea obtained using the Rev3D-OCE method. For further comparison, depth-dependent shear speed estimations using Rev3D-OCE and the Lamb wave method were put into a 2D format and superimposed on B-mode structural images of the cornea in Supplementary Fig. 1.

Figure 6a shows important differences in contrast for the elastic mapping of layers in the cornea at different IOP levels with interesting implications for diagnosis and monitoring in the ophthalmology field. Although the Lamb wave method provides an important metric that relates Lamb wave speed and IOP, the Rev3D-OCE method enables the possibility of monitoring the elasticity of single layers and how each of them are affected by the increase in IOP. In particular, the spikes in shear speed at the Bowman's membrane (zone II) and Descemet's membrane (zone V) seem to be affected differently by the IOP level, which can be of great importance for understanding ocular diseases such as glaucoma, and keratoconus, and monitoring treatments. Moreover, SE (shaded regions in Fig. 6a) at the epithelium (zone I) and endothelium (zone VI) is significantly smaller compared to

the upper (zone III) and lower (zone IV) stroma regions which is in agreement with the fact that epithelium and endothelium can be considered as more uniform tissues compared to stroma which is highly anisotropic due to the heterogeneous collagen distribution[4]. Finally, in order to test the variability of Rev3D-OCE in various healthy porcine cornea samples ($N = 10$ corneas), we reported in Fig. 6c average shear wave speed measurements per sample in every depth zone indicated in Fig. 6a. Results show a differentiated shear wave speed in every layer (zones I to VI) at lower IOP levels (10.24–15.80 mmHg) and a trend to become more uniform at higher IOPs (20.50–25.25 mmHg). More discussion of such results is conducted in the "Discussion" section.

## Discussion

The use of reverberant shear wave fields in tissue to measure mechanical properties has three key implications for the 3D OCE field and, more generally, for the study of layered media. First, the sinusoidal steady-state disturbance of the plate-shape tissue

produced by one or more sources after sufficient time (~1 s in cornea) and regardless of the initial symmetry creates diffuse fields[50] with favorable properties. For instance, it does not require prior knowledge of the direction of wave propagation for the estimation of local shear wave speed, which is a fundamental requirement for most of dynamic OCE techniques. In particular, the properties and boundary conditions of cornea such as (1) the top and bottom surfaces, (2) the heterogeneity in layer distribution along depth, and (3) the significant change in elastic properties in the cornea-sclera junction, produce multiple reflection and aberration of shear waves, which diminishes the effectivity of peak tracking techniques[51–53]. However, Rev3D-OCE does not require knowledge of a single direction of wave propagation (due to the nature of the auto-correlation in Eqs. (2) and (3)) and all wave reflections, contribute to the creation of a diffuse field.

Second, the continuous sinusoidal excitation at a given frequency promotes the propagation of a higher ratio of shear waves in tissue compared to a transient excitation in which surface acoustic waves are the predominant perturbations[49]. In elastography, propagation of shear waves is desired since shear wave speed can be directly related to shear modulus using Eq. (4). Experimental validation of Rev3D-OCE in tissue-mimicking phantoms conducted in Results Section supports this affirmation, resulting in an accuracy error of <7% in shear wave speed estimation when compared to mechanical testing. In cornea studies, most of the dynamic OCE techniques create transient waves with wavelengths comparable to the corneal thickness[35–42], which inevitably produces Lamb waves. Unfortunately, the Lamb wave speed is dependent not only on the shear modulus of the cornea, but also on the frequency of excitation and the corneal thickness, all related through a non-linear function[43,44], making it difficult to estimate shear modulus without using inverse methods and parametric models.

Third, reverberant fields created by the interaction of multiple shear waves traveling in random directions enable the detection of elasticity gradients along the depth axis, which is fundamental for detecting elasticity in corneal layers. Given the corneal boundary conditions, this key implication was validated using numerical simulations, tissue-mimicking phantom studies, and real experiments in ex vivo porcine cornea. For the numerical simulation, depth-dependent shear wave speed profiles shown in Fig. 2a-right indicate differentiated values between layers A, B, C, and D; however, reported values are slightly underestimated when compared to nominal values of shear speed provided in the material properties settings of the finite element model (maximum shear wave speed offset error of 0.44 m s$^{-1}$). This behavior can be attributed to the presence of Lamb waves and other branches of surface acoustic waves (Rayleigh waves in the top and bottom corneal surfaces). Furthermore, a minimization of those negative effects is expected due to the increase in excitation frequency and, consequently, the decrease in perturbation wavelength with respect to the corneal thickness, reducing the presence of Lamb wave modes.

For the experimental case, shear speed depth-dependent profiles obtained using Rev3D-OCE in ex vivo fresh porcine eye cornea (Fig. 4c) are qualitatively comparable with the ones reported in human cornea using atomic force microscopy[54], where spikes in Young's modulus are detected in the Bowman's layer and Descemet's membrane. Similarly, Rev3D-OCE results can be compared to other compression-based layer detection methods[22,23], where spikes in the Bowman's layer and Descemet's membrane are also detected. In Brillouin optical microscopy[31–33], and Lamb wave methods[38,41] a spike is detected near the Bowman's membrane. Finally, the passive elastography method[55] based on a broadband noise-correlation approach is able to retrieve stiffness of tissues from random displacement fields with

interesting results in the elastography of in vivo eye. The fundamental difference between Rev3D-OCE and passive elastography relies on the Rev3D use of active sources vibrating at a single frequency while producing a limiting case of 3D reverberant fields that can be modeled by Eqs. (1)–(3). Unlike Rev3D-OCE, passive elastography analyzes the broadband spectrum of noise to give estimates of wavelength based on correlation techniques and derived from time reversal principles.

In addition, as reported in other studies[39,56], corneal elasticity is anisotropic and shows differentiated wave speed along each layer. In this paper, we focus on the elastic characterization of corneal layers by averaging shear wave speed along any corneal transverse plane and sacrificing the detection of elastic anisotropies in 2D. This constitutes one of the major limitations of Rev3D-OCE, being of special importance in the detection of elastic inhomogeneities in all $x$, $y$, and $z$ axis, simultaneously, produced by ocular diseases like keratoconus. However, the SE (shaded regions) in Fig. 6a quantify the variability of shear wave speed for every depth profile subjected to a different IOP level and SE can be related to corneal anisotropy. For instance, given a IOP case, SE levels are significantly larger in regions III and IV compared to the others, corresponding to the upper and lower stroma that contain a heterogeneous distribution of collagen fibers, making stroma the main contributor of elastic anisotropy in the cornea. Finally, the repeatability study shown in Fig. 6c among $N = 10$ corneas indicates a major trend in layer elasticity becoming uniform as the cornea is stretched out by the increased level of IOP. SE can be also attributed to other sources of error such as the variability of wave speed depending on the size of the auto-correlation window with respect to the wavelength, and any swelling effect in the cornea which was not fully controlled during experiments.

A future implementation of Rev3D-OCE for in vivo, in situ, and fully non-contact method for 3D elasticity characterization of corneal layers, requires some modifications to the current system. First, the multi-pronged ring actuator can be replaced by a circular array of air-coupled focused ultrasonic transducers. This method for non-contact excitation of the cornea was successfully demonstrated by Ambrozinski et al.[42]. Subsequently, the acquisition scheme presented here can be expanded to acquire reverberant volumes in a faster fashion (time scales in the sub-second order) by upgrading the current OCT system to a faster camera with an A-line rate of 5 μs, resulting in a total acquisition time of 0.5 s. Finally, the characterization of viscoelastic parameters in cornea is of great importance[46,51,57]. Therefore, the excitation scheme of the current Rev3D-OCE method can be modified to send multi-harmonic vibrations all sent and captured simultaneously in order to calculate depth-dependent shear wave dispersion plots for the estimation of viscoelastic parameters.

## Methods

**Monte Carlo analysis.** A $20 \times 20 \times 20$ mm$^3$ field ($\Delta x = \Delta y = \Delta z = 0.1$ mm sampling resolution) in which transversal waves can only propagate at a constant phase speed of $c_s = 4$ m s$^{-1}$ is generated in MATLAB (The MathWorks, Inc. Natick, MA, USA). Inside the summation of Eq. (1), a single plane and shear wave is defined by $\hat{\mathbf{n}}_q$, $\hat{\mathbf{n}}_{ql}$, and $v_{ql}$, with the realization of four uniformly distributed random variables. First, the realization of angles $\varphi$ and $\theta$, each one covering a range of $[0, 2\pi]$ radians, for the definition of $\hat{\mathbf{n}}_q$, followed by the realization of angle $\alpha$, covering a range of $[0, 2\pi]$ radians for the definition of $\hat{\mathbf{n}}_{ql}$, and then the realization of the scalar value $v_{ql}$, covering a range of $[-v_{max}, v_{max}]$, where $v_{max} = 1$ m s$^{-1}$. The term e$^{-i\omega_0 t}$ can be factored from Eq. (1), and the wave number is defined as constant $k = \omega_0/c_s$ in the whole field with the selection of $f_0 = 2$ kHz as the temporal frequency, leading to $\omega_0 = 2\pi f_0$. A total of 10,000 realizations of all random variables, following the described order, generates a spatial and complex-valued particle velocity vector field $\mathbf{V}(\boldsymbol{\varepsilon})$, where $\mathbf{V}(\boldsymbol{\varepsilon}, t) = \mathbf{V}(\boldsymbol{\varepsilon})e^{-i\omega_0 t}$. Then, the three components of $\mathbf{V}(\boldsymbol{\varepsilon})$ are: $V_x(\boldsymbol{\varepsilon}) = \mathbf{V}(\boldsymbol{\varepsilon}) \cdot \hat{\mathbf{e}}_x$, $V_y(\boldsymbol{\varepsilon}) = \mathbf{V}(\boldsymbol{\varepsilon}) \cdot \hat{\mathbf{e}}_y$, and $V_z(\boldsymbol{\varepsilon}) = \mathbf{V}(\boldsymbol{\varepsilon}) \cdot \hat{\mathbf{e}}_z$, where $\hat{\mathbf{e}}_x$, $\hat{\mathbf{e}}_y$, and $\hat{\mathbf{e}}_z$ are unit vectors in the $x$, $y$, $z$ directions, respectively. Since in OCT the motion is measured in the $z$-axis, $V_z(\boldsymbol{\varepsilon})$ is the spatial field of interest. Figure 1b shows the real

part of $V_z(\mathbf{\varepsilon}, t_0)$, defined as $\text{Real}\{V_z(\mathbf{\varepsilon}, t_0)\} = \text{Real}\{V_z(\mathbf{\varepsilon})e^{-i\omega_0 t_0}\}$, for an arbitrary instant $t_0$. Finally, the real part of the complex-valued 2D auto-correlation maps taken from $V_z(\mathbf{\varepsilon})$ are calculated in the following planes: (1) $xy$-plane, (2) $xz$-plane, and (3) $yz$-plane as shown in Fig. 1c. Profile plots taken along each axis ($\Delta x$, $\Delta y$, $\Delta z$) are compared to Eqs. (2) and (3) for further validation.

**Numerical simulations**. Numerical simulations using finite elements in Abaqus/CAE version 6.14-1 (Dassault Systems, Vélizy-Villacoublay, France) are conducted in a four-layered model of cornea for two dynamic cases: (1) sinusoidal steady-state using harmonic excitation, and (2) transient tone burst excitation. The cornea model consists of a 3D solid deformable part formed by the vertical composition of four curved layers—A, B, C, and D—each approximately 0.25 mm thick and 10 mm in diameter. The 3D solid was meshed using 140770 hybrid, quadratic and tetrahedral elements (C3D10H). Linear elastic material properties were chosen for each section or layer using Young's modulus of $E_A = 18.75$ kPa, $E_B = 12$ kPa, $E_C = 5.07$ kPa, and $E_D = 1.92$ kPa, where the subscript denotes each layer. The Young's modulus selection and distribution of layers from stiffer (A) to softer (D) were based on the decreasing depth-dependent wave speed tendency reported in Wang et al.[38] for porcine cornea. In all cases, the density was selected to be $\rho = 1000$ kg m$^{-3}$—similar to water—and the Poisson's ratio was $\nu = 0.49999$, near incompressibility. Since the material properties of each section are homogeneous and isotropic, then, for $\nu \approx 0.5$, the relationship between shear (G) and Young's modulus (E) can be approximated as $E \approx 3G$[30]. The equivalent shear wave speed for each section is calculated using Eq. (4): $c_s^A = 2.5$ m s$^{-1}$, $c_s^B = 2$ m s$^{-1}$, $c_s^C = 1.3$ m s$^{-1}$, and $c_s^D = 0.8$ m s$^{-1}$, where the superscript denotes each layer. The outer border of the cornea model, simulating the limbus, is subjected to encastre boundary conditions (zero displacement and rotation).

For the sinusoidal steady-state analysis, the outer surface of layer A is subjected to eight body force loads distributed circularly around the cornea (see Fig. 2a-left). The simulation was selected to be steady-state dynamic direct for a vibration frequency of 2 kHz. After the simulation, the 3D complex-valued particle velocity field along the $z$-axis is extracted and subsequently processed using auto-correlation and fitted to Eq. (2) for the depth-dependent shear wave speed estimation of each layer (see Fig. 2a-right). For the transient tone burst analysis, the central part of the outer surface of layer A is subjected to a Gaussian distribution body force with $\sigma = 03$ mm along the symmetry line for 1 ms (see Fig. 2b-left). The type of simulation was selected to be dynamic-implicit for 4 ms in order to let the Lamb wave propagate along the medium without producing reflections from the outer boundaries. After the simulation, the 3D particle velocity field along the $z$-axis is extracted for each time instant ($\Delta t = 10$ μs) during the simulation period. Subsequently, space-time map representations (propagation path following the corneal curvature along $x$-axis vs. time) are extracted for each layer and processed using a 2D Fourier transform for dispersion analysis (Lamb wave speed vs. frequency plots as shown in Fig. 2b-right). We have also included speed dispersion plots when a 0.5 ms transient force (same Gaussian spatial extent $= 0.3$ mm) is applied to the corneal model in Fig. 2b-left. Finally, the frequency component of 1 kHz was selected for the calculation of Lamb wave speed vs. corneal depth (Lamb wave method). This plot is compared against shear wave speed vs. corneal depth results calculated in the sinusoidal steady-state case in Fig. 2a-right.

**Sample preparation**. Tissue-mimicking phantom layers were made of gelatin powder of different concentrations for simulating softer (3% gelatin) and stiffer (5% gelatin) tissues. In all cases, the same concentration of 0.5% intralipid powder was used to provide optical scattering to the phantoms (measured refractive index of phantoms ~1.35). Both layers were vertically distributed as top (3%) and bottom (5%). The average thickness of the top layer was measured to be ~0.3 mm and the bottom layer is considered as a semi-infinite medium. The approximate size of each phantom was $70 \times 70$ mm$^2$ in the lateral extent, and 30 mm along depth. The Young's modulus of each phantom concentration was measured by taking three cylindrical samples ($N = 3$) of ~40 mm in diameter, and conducting a stress–relaxation compression test in a MTS Q-test/5 universal testing machine (MTS, Eden Prairie, Minnesota, USA) at a strain of 5% for a total time of 600 s. The stress-time plots obtained were fitted to a Kelvin-Voigt fractional derivative rheological model for the frequency-dependent Young's modulus ($E$) estimation[58]. For the excitation frequency used in the experiments (2 kHz), Young's moduli were found to be: $E_{3\%}^{MM} = 3.97 \pm 0.83$ kPa and $E_{5\%}^{MM} = 12.36 \pm 1.71$ kPa, for 3 and 5% concentrations, respectively. Assuming that the phantom is an isotropic and homogeneous material of density of ~1 g cm$^{-3}$, and a Poisson's ratio of ~0.5, the shear wave speed can be calculated[30] and reported as: $c_{3\%}^{MM} = 1.15 \pm 0.12$ m s$^{-1}$, and $c_{5\%}^{MM} = 2.03 \pm 0.14$ m s$^{-1}$, for 3% and 5% concentrations, respectively.

Ten freshly enucleated and non-scalded porcine eyeballs were obtained from an abattoir (Joe's Meat Market, Ontario, NY, USA) with all experiments being performed within 6 h of their collection and following ethical regulations approved by University of Rochester. Only intact eyes with clear corneas were used in this study. After removal of extra-ocular muscle and fat tissue, the eyeballs were washed in phosphate buffered saline (PBS) and placed in a custom-built eye holder. A needle (outer diameter of 0.9 mm) penetrated the eyeball laterally through the sclera (3 mm from the limbus) until reaching the anterior chamber for IOP control during experiments. The needle is tubed to a syringe with PBS and a digital

pressure gage (model DPGWB-04, Dwyer Instruments Inc., Michigan City, Indiana, USA) through a three-way stopcock, which is used for increasing the IOP level in the anterior chamber. Four average IOP levels were set in the eyeball: 10.24, 15.80, 20.50, and 25.25 mmHg. The maximum IOP deviation from the average level was measured to be 1.12 mmHg during the acquisition time (60 s). Each cornea was irrigated with PBS every minute to avoid dehydration effects.

**Experimental setup**. The experimental setup consists in a spectral-domain PhS-OCT system for the motion detection of waves generated in the sample by a synchronized mechanical excitation system (Supplementary Fig. 2). The mechanical excitation system begins with a function generator (AFG320, Tektronix, Beaverton, OR, USA) output signal connected to an ultra-low noise power amplifier (PDu150, PiezoDrive, Callaghan, NSW, Australia) feeding a piezoelectric bender poled in a parallel configuration of $10 \times 45$ mm$^2$ surface dimensions (BA4510, PiezoDrive, Callaghan, NSW, Australia). A 3D printed ring containing eight vertical equidistant and circular-distributed rods is attached to one of the ends of the piezoelectric bender. The rods are slightly touching the corneal epithelium in a concentric configuration and produce reverberant fields in cornea when the piezoelectric bender is excited. The ring shape allows the imaging of the cornea using the OCT system, while the rods introduce mechanical excitation. The rod tips were polished to avoid damaging the cornea. The signal set in the function generator was a 2 kHz harmonic and continuous sinusoidal function running freely during the entire acquisition. For the generation of Lamb waves in cornea, the printed ring was replaced by a single 3 cm vertical rod with a 1 mm diameter metal sphere attached to its end. The sphere was slightly and normally touching the corneal epithelium at a position parallel to one of the region of interest (ROI) axes crossing through the center of the cornea. The signal set in the function generator for this case was one cycle of a 1 kHz tone (1 ms tone) burst every time the acquisition system sends a trigger signal.

The selection of the excitation frequency (2 kHz), the number of tips in the multi-pronged excitation ring (between 6 and 8), and the excitation time needed to reach the diffuse steady-state regime (~1 s) was guided by results from finite element simulations of porcine cornea. We have chosen the 2 kHz excitation as the intermediate case, balancing the desire for smaller wavelengths while avoiding stronger attenuation of waves below the sensitivity floor for motion detection using the current PhS-OCT system. Moreover, the multi-pronged ring was 3D-printed in a single piece attached to the piezoelectric actuator; therefore, each tip produces waves in tissue with the same temporal phase. On one hand, adding more tips along the circular ring and exciting them with the same frequency and phase will result in producing circular waves as if produced by a continuous ring instead of a collection of tips (following the Huygens-Fresnel principle[59]). On the other hand, using fewer tips in the ring will result in only a few wave sources, therefore, reducing the probability of generating a multidirectional reverberant field. Quantitatively speaking, we have determined in simulations and confirmed in experiments that between 6 and 8 tips excited at 2 kHz in porcine cornea generates a wave field with the desired reverberant properties. Finally, during experiments, we produced a continuous harmonic excitation of 2 kHz, ensuring steady state for 1 s before the imaging is activated, and confirmed the steady-state regime experimentally by analyzing the spatial wave field that turns out to be periodic, allowing its representation by a single amplitude and a phase parameter for every single pixel in the corneal volume.

For motion detection, a custom-built spectral-domain PhS-OCT system was implemented. The OCT system is spectrometer-based, using a light source composed of a super-luminescent light emitting diode (EXS210045-01, EXALOS, Switzerland) with a central wavelength at about 1307 nm and the FWHM spectral width of about 100 nm. The average output power was about 12 mW. The light source was split (50:50 power split ratio) and delivered to the sample arm, consisting of dual-axes galvanometric mirrors, and to an adjustable reference arm. The collected back-scattered light from the sample is combined with the reference light beam and sent to a custom-built spectrometer (more details of the system in Supplementary Methods). The lateral resolution of the system is measured to be 15 μm, and the axial resolution, measured from the FWHM of the depth profiles, is between 15 and 20 μm in air. The A-line acquisition rate of the system is 25 kHz (temporal sampling periodicity of $T_s = 40$ μs). ROIs in the gelatin phantom sample and porcine cornea samples were defined to be $1 \times 1$ mm$^2$, and $8 \times 8$ mm$^2$, respectively.

**Data acquisition**. The MB-mode acquisition approach[51], is used in the quasi-synchronized (for Rev3D-OCE) and fully-synchronized (for Lamb wave method) versions. For the Lamb wave method, the acquisition consists of triggering the excitation (1 ms tone) and acquiring $M = 100$ A-lines during $M \times T_s = 4$ ms at a single ($x_0$, $y_0$) position in cornea along depth. Subsequently, the galvanometer controlling $x$-axis changes its location to the consecutive ($x_0 + \Delta x$, $y_0$) position, and the excitation and acquisition is repeated. When $x$-axis coverage of 8 mm is reached, then the galvanometer controlling the $y$-axis moves to the next ($x_0$, $y_0 + \Delta y$) position and the $x$-scanning process is repeated. The process is stopped when the $y$-axis coverage of 8 mm is reached. The number of samples taken in the $x$ and $y$ axis of the ROI is $100 \times 100$ ($\Delta x = \Delta y = 80$ μm). The total acquisition time was 60 s per experiment. Reorganizing the 4D data (3D spatial, 1D temporal) into 100 3D spatial volumes of $8 \times 8 \times 5$ mm$^3$, where each volume corresponds to a virtual simultaneous snapshot of the cornea motion at one time instant ($T_s = 40$ μs)

covering 4 ms, the apparent volumetric frame rate is 25 kHz. For the Rev3D-OCE approach, the excitation is not triggered by the acquisition at every single $(x_0, y_0)$ position. Instead, the excitation is initially burst at 2 kHz harmonic sinusoidal signal running freely. Approximately 1 s later, after reaching the motion diffuse regime in cornea, the OCT system initiates the acquisition by recording $M = 100$ repetitions at any $(x_0, y_0)$ position and moving to the next $(x_0, y_0 + \Delta y)$ position every 6 ms ($M \times T_s + 2$ ms). Therefore, we ensure phase synchronization by propagating precisely 12 cycles of the 2 kHz harmonic signal every exact 6 ms. The calculation of particle velocity from two consecutive A-lines is implemented using the Loupas et al.[60] approach for increasing the signal-to-noise ratio. As a result, the motion signal extracted from the cornea at any $(x_0, y_0)$ position is a real-valued time domain signal representing particle velocity in the z-axis.

**Data processing for Rev3D-OCE.** In Rev3D-OCE, the 4D scalar field $V_z^{\mathrm{OCT}}(\mathbf{\epsilon}, t)$ (3D spatial, 1D temporal) from cornea acquired using the PhS-OCT system is represented as a temporal sequence of 3D particle velocity volumes. This field is a real-valued scalar field that can be separated as:

$$V_z^{\mathrm{OCT}}(\mathbf{\epsilon}, t) = \mathrm{Real}\{V_z(\mathbf{\epsilon})e^{-i\omega_0 t}\} = A(\mathbf{\epsilon})\cos(\Phi(\mathbf{\epsilon}) - \omega_0 t), \quad (6)$$

where the complex-valued spatial particle velocity field in cornea $V_z(\mathbf{\epsilon}) = A(\mathbf{\epsilon})e^{i\Phi(\mathbf{\epsilon})}$ can be represented with magnitude $A(\mathbf{\epsilon})$ and phase $\Phi(\mathbf{\epsilon})$, and $e^{-i\omega_0 t}$ is the complex representation of the steady-state harmonic time domain vibration. This interpretation comes from Eq. (1), where the time harmonic term in the exponential can be factored from the summation. The real part operator is set in Eq. (6) because the acquisition system measures real-valued motion signals. Then, the goal is to recover a version of $V_z(\mathbf{\epsilon})$ from $V_z^{\mathrm{OCT}}(\mathbf{\epsilon}, t)$, since the former contains magnitude and phase information of the reverberant field needed for the calculation of auto-correlation according to Eqs. (2) and (3).

The processing pipeline for the Rev3D-OCE method is shown in Supplementary Fig. 3. First, the B-mode version of the acquired 3D volume is used for the estimation of the top (epithelium) and bottom (endothelium) topographic surfaces using intensity peak detection techniques and thresholding methods. Three hundred intermediate surfaces between top and bottom layers, $S_j$ for $j = 0, 1, 2, ..., 299$, are estimated using cubic interpolation to account for irregularities and anisotropic variations of corneal thickness. Then, the spatial domain of $V_z^{\mathrm{OCT}}(\mathbf{\epsilon}, t)$ is converted into a 3D point cloud, and points not belonging to the cornea boundaries are eliminated. Subsequently, particle velocity values are projected to a given topographic surface $S_{j0}$ from $V_z^{\mathrm{OCT}}(\mathbf{\epsilon}, t_0)$ using interpolation for each time instant $t_0$, compensated for the corneal curvature, and stored in a 3D matrix representation (spatial 2D, temporal 1D). For a given instant $t_0$, a profile on the 3D matrix corresponds to a motion frame of the corneal surface been analyzed. Each motion frame is filtered using a 2D spatial isotropic band-pass filter kernel. The filter was designed in the k-space domain as a donut-shaped ring with Gaussian borders for the elimination of wavelengths values in $V_z^{\mathrm{OCT}}(\mathbf{\epsilon}, t_0)$ not supported by soft tissue (Supplementary Fig. 3). Based on a number of studies conducted in porcine corneas[25,36,38,42,61] at different excitation frequencies and IOP levels, we have defined a range of wave speeds supported by such tissue from 0.2 m s$^{-1}$ to 10 m s$^{-1}$ enabling the determination of the upper and lower limits of the spatial filter. Subsequently, for a given position $\epsilon_0$ in $V_z^{\mathrm{OCT}}(\mathbf{\epsilon}_0, t)$ the temporal signal is filtered using a narrow finite input response (FIR) filter centered at 2 kHz in the frequency domain. Both spatial and temporal filtering processes provide a new 3D representation named $\tilde{V}_z^{\mathrm{OCT}}(\mathbf{\epsilon}, t)$.

A temporal Fourier transform $\Im\{\cdot\}$ is applied to each spatial position in the 3D matrix $\tilde{V}_z^{\mathrm{OCT}}(\mathbf{\epsilon}_0, t)$ along the time extension, where a clear spike is centered at $f_0 = 2$ kHz (Supplementary Fig. 3). Formally, this procedure is equivalent to taking the Fourier transform to Eq. (6) as:

$$\Im\{V_z^{\mathrm{OCT}}(\mathbf{\epsilon}, t)\} = A(\mathbf{\epsilon})\sqrt{\frac{\pi}{2}}\left[e^{-i\Phi(\mathbf{\epsilon})}\delta(\omega + 2\pi f_0) + e^{i\Phi(\mathbf{\epsilon})}\delta(\omega - 2\pi f_0)\right], \quad (7)$$

where $\delta(\omega \pm 2\pi f_0)$ is the Dirac's delta centered in $\omega = \mp 2\pi f_0$ respectively. Therefore, the magnitude $A(\mathbf{\epsilon})\sqrt{\pi/2}$ and phase $\Phi(\epsilon)$ calculated from the temporal Fourier transform of the 3D matrix is equivalent to $\Im\{V_z^{\mathrm{OCT}}(\mathbf{\epsilon}, t)\}$ evaluated at $f_0$ in the topographic surface $S_{j0}$ ($\mathbf{\epsilon} \in S_{j0}$). Then $V_z(\epsilon)$, for $\mathbf{\epsilon} \in S_{j0}$, can be reconstructed as $V_z(\mathbf{\epsilon}) = \sqrt{\pi/2}A(\mathbf{\epsilon})e^{i\Phi(\mathbf{\epsilon})}$, a complex-valued 2D spatial particle velocity matrix of the cornea at surface $S_{j0}$. This procedure can be iterated for every surface $S_j$ within the cornea until reaching $S_{299}$, which corresponds to the corneal endothelium.

Subsequently, a 2D complex-valued auto-correlation map is obtained from a $2 \times 2$ mm$^2$ windowed region of $V_z(\mathbf{\epsilon})$. The average of auto-correlation curves ($N = 360$ curves) taken along radial cuts of the 2D auto-correlation map covering 360 degrees is fitted to Eq. (2) since the particle velocity is perpendicular to the correlation direction. Local estimated values of wave number $k^*$ are converted into shear wave speed using $c_s = \omega_0/k^*$, given the excitation frequency $\omega_0 = 2\pi f_0$. This process is repeated for all windows covering the whole ROI in $V_z(\mathbf{\epsilon})$ for a given surface $S_{j0}$ and generating a 2D shear wave speed map $c_s(\mathbf{\epsilon})$. Then, each $c_s(\mathbf{\epsilon})$ is generated for every surface within the cornea and stacked in a 3D matrix. Finally, the B-mode 3D volume of cornea is fused with the generated 3D shear wave speed map for visualization purposes. Depth-dependent shear wave speed maps and plots can be extracted by sectioning the 3D matrix with a transversal plane or line parallel to the z-axis, respectively.

The impact of the 2D auto-correlation window size on the speed estimations of the cornea was explored in Supplementary Fig. 4 in which a motion frame extracted at the corneal depth $z_0 = 0.30$ mm (Supplementary Fig. 4a) was analyzed using different window sizes. As shown for a $4 \times 4$ mm$^2$ window (Supplementary Fig. 4b), auto-correlation cuts extracted at different angles are not identical, mainly attributed to the anisotropic mechanical properties of corneal layers (and the main source of SE in Fig. 4b), in contrast to results in gelatin phantoms that are highly isotropic. Moreover, the auto-correlation window selection can impact the accuracy of speed estimations as shown in Supplementary Fig. 4c when compared to Eq. (2). Larger windows produce more accurate and repeatable speed estimation, at the cost of losing lateral elastography resolution (Supplementary Fig. 4d–f).

**Data processing for the Lamb wave method.** In the Lamb wave method, Eqs. (6) and (7) are not valid since a transient excitation (particle velocity limited in space and time) produces a cylindrical and axisymmetric Lamb wave to propagate outward in the corneal tissue. The processing pipeline for the Rev3D-OCE method is shown in Supplementary Fig. 5. The analysis is performed in $V_z^{\mathrm{OCT}}(\mathbf{\epsilon}, t)$ for every surface $S_j$, where $j = 0, 1, 2, ..., 299$, defining a direction of wave propagation $\hat{\mathbf{r}}$. Then, for a given surface $S_{j0}$, 2D space-time maps are generated from $V_z^{\mathrm{OCT}}(\mathbf{\epsilon}, t)$ along the $\hat{\mathbf{r}}$-$t$ space. Subsequently, a 2D Fourier transform is applied to the space-time map and is converted into the $k$-$\omega$ space. For every $\omega$, the wave number of higher intensity, $k^*(\omega)$, is calculated using peak detection methods. Frequency samples $[\omega, k^*(\omega)]$ with low signal-to-noise ratio are discarded during the estimation process. Wave number is converted into Lamb wave speed using $c_L(\omega) = \omega/k^*(\omega)$ and plotted versus $\omega/2\pi$; the results are known as dispersion plots. The form of the curve will determine if the Lamb wave mode is symmetric or antisymmetric. Dispersion plots $c_L(\omega)$ are calculated for every surface $S_j$ within the cornea until reaching $S_{299}$, which corresponds to the corneal endothelium. Depth-dependent Lamb wave speed plots can be extracted by choosing an excitation frequency $\omega_0 = 2\pi f_0$.

## Data availability

All relevant data are available from the authors. A reporting summary for this article is available as a Supplementary Information file.

## Code availability

All relevant codes are available from the authors.

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

## Acknowledgements

The instrumentation engineering development for this research benefited from support of the II-VI Foundation. This work was supported by the National Institute of Health (NIH) (R21EB025290). Fernando Zvietcovich was supported by the Fondo para la Innovacion, la Ciencia y la Tecnologia FINCyT—Peru (097-FINCyT-BDE-2014). We would like to acknowledge the support of the Hajim School of Engineering and Applied Sciences at the University of Rochester and the Center of Excellence in Advanced Functional Materials at Suranaree University of Technology. We would like to thank Linda Weidman for assistance with the manuscript editorial corrections.

## Author contributions

F.Z., K.J.P., J.P.R., and P.M. conceived and guided the project; F.Z and K.J.P designed the experiments; F.Z. and P.P. performed the experiments; P.P. and P.M. built and adapted the OCT system for elastography; F.Z. processed the data and conducted the analysis; F.Z., K.J.P., J.P.R., and P.M. wrote and edited the manuscript.

## Competing interests

The authors declare no competing interests.
