## [Peer Review File · Nature Communications]

Reviewers' comments:

Reviewer #1 (Remarks to the Author):

In this manuscript, the authors describe and implement a new technique for measuring the depth-wise stiffness of a cornea using a technique called reverberant optical coherence elastography. The authors provide a theoretical foundation for the technique, Monte Carlo simulations, finite element modeling, and comparison with transient Lamb wave measurements. While the technique is impressive, and the results are laudable, there are some concerns before this manuscript can be accepted for publication.

The biggest limitation of this manuscript is the small sample size and lack of pathological assessments. For a journal of this caliber, additional samples and pathological measurements are needed in order to demonstrate the efficacy and utility of this technique.

Why was the 2mm x 2mm window chosen? Is this window dependent on the shear speed? If then, is there some a priori information needed before the data can be processed? Does one have to know the speed of the wave beforehand in order for a sufficient fitting window? Does the fitting window size have any effect on the results?

I think it is unfair to compare 1 kHz in the Lamb wave technique to 2 kHz excitation with the reverberant technique. The authors should address this with additional simulations and/or experiments.

The fitting of the autocorrelation in the corneal results is unusually clean for experiential data. Can authors demonstrate repeatability of such fitting?

Moreover, simulations and measurements on layered phantoms must be demonstrated.

What is the transverse elasticity resolution?

How were the frequency limits in the Lamb wave technique chosen?

Why was an excitation frequency of 2kHz chosen?

Why did the excitation method have 8 points? Why not 4 or 16?

How much time is sufficient for the system to reach "steady state"?

The manuscript needs more details about the excitation and imaging setup. Was there a new synchronized excitation for each M-mode image? Or was there continuous excitation and subsequent M-B-mode imaging?

The authors should address the effects of elastic anisotropy.

I would like to see additional discussion on passive elastography and the difference between reverberant shear waves and standing waves.

In the derivation in the supplemental materials, α is not defined. The authors also derive based on x , but then change the equations to z (the optical axis). I think that it would be much more straightforward if the derivations were performed with respect to z with a small preface that z is the optical axis with respect to the OCE system.

The cornea was irrigated with 1X PBS. PBS is not isotonic with the cornea and will cause the cornea to swell. The authors should address this and any effects of cornea edema on their results.

In the Monte Carlo simulations, what do the standard deviations represent?

How did the authors determine which wavelengths are not supported by the tissue in the data processing?

What are the major limitations of this technique?

There are some minor grammatical and typographical errors
"being the stroma is the thickest layer in the cornea"
"the needle is tubbed"

Reviewer #3 (Remarks to the Author):

This manuscript demonstrates an optical coherence elastography (OCE) technique to measure the shear moduli of different layers in the cornea. This technique involves exciting reverberant acoustic waves in the cornea using several piezoelectrically vibrating tips around the limbus and imaging the speckle-like interference pattern of the waves using phase-sensitive optical coherence tomography. From a 2D spatial auto-correlation analysis of the acoustic displacement field at a specific frequency, the acoustic wavelength of each layer is calculated, from which shear acoustic speed and modulus is determined. The general principle of this technique has been known. However, this manuscript reports its first application in conjunction with optical imaging. The novelty of this work is high.

In terms of significance, the ability to measure the elastic moduli of different layers in the cornea with high accuracy is nontrivial and has potential to be clinically useful for the assessment of corneal health and diseases, as well as refractive surgeries. Having said that, however, it is unclear to me whether the technique indeed allows for accurate, high-resolution elastography with practical advantages over alternative techniques such as ultrasound-excited shear wave or lamb wave OCE. The manuscript is lacking in careful quantitative validation of the technique, and there are reasons to be concerned that a part, if not the majority, of the depth-dependent features they observe may not be real and inaccurate.

The key assumption is that acoustic waves are randomly mixed up and reach the so-called "diffuse" regime, where their speeds are close to the pure-shear speed. This important assumption needs to be verified experimentally. The authors may have different ideas how to validate. Perhaps they can start with a circular plate made of uniform material and then use corneal phantoms consisting of multiple layers of known elastic moduli. It should be noted that the effects of the acoustic attenuation and the boundary conditions also influence the quality of the reverberation pattern.

Some specific questions are:

1. The ideal diffuse regime requires acoustic waves in all directions to be excited with relatively equal magnitudes. To what extent was this state achieved? Show that the result is independent of the excitation geometry. Vary the phases and locations of the excitation points and show that the results (depth profiles) are consistent.
2. The technique analyzes layer by layer. Therefore, any noise and artifacts will show up as a depth profile. Quantitative validation with known samples and phantoms is needed.
3. Likewise, any radial variation of a sample can affect the propagation of waves, and this can show up as depth dependence. To understand to what extent they contain artifacts, experiments

with a layered phantom will be very helpful.

4. In Fig. 2, it seems a constant speed was assumed in the FEM simulation, rather than simulating the generation of waves from the PZT contacts.
5. Show raw data $B(x,y)$ measured with the cornea. The ROI in the sample was defined to be 8×8 mm. Analyze B for smaller ROI in x, y . How uniform is the result across the cornea?
6. Does the spatial correlation along the x -axis versus y -axis produce the identical result?
7. Show measurement repeatability for the same cornea.
8. Show data for several corneas with similar sizes and show inter-sample variability.
9. When each PZT excites Lamb wave (primarily lowest-order antisymmetric), how can 8 PZT end up generating random waves? Or, is the technique measuring interference of 8 Lamb waves?
10. The acoustic attenuation at 2 kHz is non-negligible. It may be that each wave makes no more than one reflection from the edges of the cornea and the extent of mode conversion may be small. To what extent the diffuse field assumption valid?
11. Please explain "Furthermore, if a transverse isotropic model of the material is assumed, G corresponds to out-of-plane shear modulus."
12. Is the method independent of sample thickness? Verify using phantoms of uniform material but different thicknesses.
13. The current method does not provide any transverse resolution. Reducing the size of ROI will decrease the quantitative accuracy. Keratoconus is usually a focal disease that may have mechanical inhomogeneity in both x, y , and z . This may be a limitation of this technique.
14. How smaller feature in z can be measured? At 2 kHz and $c_s = 1$ m/s, the acoustic wavelength 2 mm, almost two orders of magnitude larger than the thickness of membrane in the cornea. Although the acoustic wave is not directly translated to the spatial resolution of elastography, a controlled experiment using a phantom containing a thin layer with various thickness can demonstrate spatial resolution in z .

August 9, 2019

Dear reviewers

Many thanks to you and for the helpful comments. The manuscript has been improved substantially by the suggested changes and insightful comments. All changes made from the original version have been highlighted in red in the new manuscript file.

REVIEWER #1

In this manuscript, the authors describe and implement a new technique for measuring the depth-wise stiffness of a cornea using a technique called reverberant optical coherence elastography. The authors provide a theoretical foundation for the technique, Monte Carlo simulations, finite element modeling, and comparison with transient Lamb wave measurements. While the technique is impressive, and the results are laudable, there are some concerns before this manuscript can be accepted for publication. The biggest limitation of this manuscript is the small sample size and lack of pathological assessments. For a journal of this caliber, additional samples and pathological measurements are needed in order to demonstrate the efficacy and utility of this technique.

We have added more cases of normal porcine cornea (sample number = 10) in order to expand the analysis of the proposed technique and test the repeatability of the layer detection in different tissue conditions (subjected to different IOPs). Those cases were added in Section 2.5 and Figure 5 as suggested by the reviewer. We also characterized the axial elastography resolution of our technique using a layered phantom which we added in Section 2.3 and Figure 3.

Why was the 2mm x 2mm window chosen? Is this window dependent on the shear speed? If then, is there some a priori information needed before the data can be processed? Does one have to know the speed of the wave beforehand in order for a sufficient fitting window? Does the fitting window size have any effect on the results?

The window size must contain at least half a wavelength of the mechanical wave for a confident estimation of local wavelength, and consequently, shear wave speed. Therefore, the window size for healthy porcine cornea for all IOP cases was chosen by preprocessing the volume wave displacement field and making sure to satisfy the wavelength condition. We found that a 2×2 mm window size satisfies all cases. *A priori* knowledge of shear wave speed is not strictly necessary for the window selection, only an approximate lower bound. We have included this comment at the beginning of the second paragraph of Section 2.4. We also expand the analysis on the effect of the window size with respect to the excitation wavelength in the estimation of shear wave speed at the end of the first paragraph of Section 2.4 and in the Supplementary Figure 1.

I think it is unfair to compare 1 kHz in the Lamb wave technique to 2 kHz excitation with the reverberant technique. The authors should address this with additional simulations and/or experiments.

We re-ran the simulations applying an axisymmetric Gaussian ($\sigma = 0.3$ mm) transient ($t = 0.5$ ms) force in the center of the layered cornea model using the Lamb wave method (equivalent to 2 kHz). From the simulation results, we didn't observe a major improvement in the layer discrimination contrast due to the nature of the transient propagation of waves in thin layers, which is basically dominated by the Lamb wave effect. We also confirmed the same antisymmetric zero order tendency which reinforces the evidence of Lamb wave propagation. We included these results in Section 2.2. Based on this finding, we decided to conduct no further Lamb wave experiments in porcine cornea at the 2 kHz excitation. Further experimental evidence of Lamb wave propagating at similar and higher temporal frequencies is found in Reference [62].

The fitting of the autocorrelation in the corneal results is unusually clean for experiential data. Can authors demonstrate repeatability of such fitting?

The autocorrelation curve was obtained after 2D spatial filtering of the wave pattern which partially explains the smoothness of the curve. The 2D spatial filtering suppresses most of the noise at high frequencies preserving the spectral range of possible wave numbers of shear waves propagating in the cornea. In addition, the auto-correlation operation is analogous to a moving average filtering process, which further helps to clean the curve. Auto-correlation curves in Figures 3, 4b, and Supplementary Figure 1 were modified to be the average of auto-correlation curves ($n = 360$ curves) taken along radial cuts of the 2D auto-correlation plot covering 360 degrees. The standard error (SE) is also shown within the plots as shadowed regions.

Moreover, simulations and measurements on layered phantoms must be demonstrated.

We included experimental results of reverberant fields in a dual layered phantom for the spatial elastography resolution characterization along the depth axis in Section 2.3. With respect to simulations, we believe Section 2.2 shows convincing results on the capabilities of the proposed technique in detecting layers in a corneal-shaped media with 4 layers. A more extended study of the elastography resolution of the proposed technique in simulations and experiments with phantoms goes beyond the scope of this paper; however, we consider the topic to be very important and it will be the subject of a future publication.

What is the transverse elasticity resolution?

We found an axial elastography resolution of $\delta_z = 55.5 \mu\text{m} \pm 17.3 \mu\text{m}$, measured along the average transition from the softer top (3% gelatin concentration) to stiffer bottom (5% gelatin concentration) phantom layers distributed as shown in Figure 3. We included this analysis in Section 2.3.

How were the frequency limits in the Lamb wave technique chosen?

In Figure 6a, we analyzed the speed dispersion of the Lamb wave discriminating frequency samples $[\omega, k^*(\omega)]$ with low signal-to-noise ratio due to the attenuation effect. We added this comment in the second paragraph of Section 2.5 and in the fifth paragraph of Data Processing Scheme in Section 4. As it happens in viscoelastic tissues such as

porcine cornea, attenuation of waves increases when excitation frequency is also increased which explains a cut-off frequency in every curve. However, when the IOP is increased in cornea, the tissue becomes rigid (an increase of store modulus with respect to loss modulus) and the attenuation (related to the imaginary part of the wave number) decreases, allowing propagation of Lamb waves for higher excitation frequencies as shown in Figure 6a.

Why was an excitation frequency of 2kHz chosen?

We have chosen 2 kHz as a trade-off between producing smaller excitation wavelengths (fundamental for producing multiple reflections in both boundaries of the cornea and avoiding Lamb waves to be the predominant perturbation), and avoiding stronger attenuation of waves below the sensitivity floor for motion detection using the current PhS-OCT system. This has been added at the fourth paragraph of Section 3. In addition, the time sampling resolution of the PhS-OCT system is $T_s = 40\mu\text{s}$, which allows at most ~ 12 sampling points along one cycle of the 2 kHz harmonic signal. Therefore, increasing the frequency would diminish the measurement of the wave field as it gets closer to the Nyquist limit. Upgrading the current OCT system to have higher time, motion, and space resolution capabilities will enable to work at much higher frequencies and may result in increased elastography resolution.

Why did the excitation method have 8 points? Why not 4 or 16?

In this paper experiments, the multi-pronged ring was 3D-printed in a single piece attached to the piezoelectric actuator; therefore, each tip produces waves in tissue with the same temporal phase. In theory, adding more tips along the circular ring and exciting them all with the same frequency and phase will converge in producing circular waves as it was produced by a continuous ring instead of a collection of tips following the Huygens-Fresnel principle. On the other hand, using fewer tips in the ring will produce fewer wave sources, eventually reducing the probability of generating a uniform reverberant field (which is dependent on the number of wave sources and the reflections produced in the boundary conditions). Quantitatively speaking, we have determined in simulations and confirmed in experiments that between 6 to 8 tips excited at 2 kHz in porcine cornea generates a wave field in between the two undesired limit cases: producing ring waves, and not generating enough reflections to create a reverberant field. We have recently discovered that we can make the number of tips independent to the upper limit case by making each tip have a random and different temporal phase by adding a single piezo-electric actuator to each tip and avoiding ring waves. This is commented on in the fourth paragraph of Section 3 and will be a matter of a subsequent publication.

How much time is sufficient for the system to reach “steady state”?

A good criterion for steady state condition is to consider an excitation time equivalent to, at least, three theoretical round trips of the wave from one extreme location to the opposite extreme within the ROI in the tissue. Using the lower bound of $c_s = 1$ m/s, the upper bound of maximum corner-to-corner distance of 10 mm, and 3 round trips of the wave, the required time to reach steady estate will be $(10 \text{ mm})/(1 \text{ m/s}) \times 6 = 60 \text{ ms}$. Therefore, the time limit to reach steady state is dependent on the excitation frequency,

tissue elasticity properties, and tissue boundary conditions. Experimentally, we produce a continuous harmonic excitation of 2 kHz ensuring steady state before the imaging is activated (OCT acquisition is enabled 1 second after the excitation is activated). In addition, we confirmed the steady state regime experimentally by analyzing the spatial wave field that turns out to be periodic, which allows its representation by a single amplitude and a phase value for every single pixel in the corneal volume. This discussion has been added in the fourth paragraph of Section 3.

The manuscript needs more details about the excitation and imaging setup. Was there a new synchronized excitation for each M-mode image? Or was there continuous excitation and subsequent M-B-mode imaging?

We expanded the explanation regarding the synchronization between excitation and imaging in Experimental Setup and Data Acquisition in Section 4. For the Rev3D-OCE approach, the excitation is not triggered by the acquisition at every single (x_0, y_0) position. On the contrary, the excitation is initially burst at 2 kHz harmonic sinusoidal signal running freely while the OCT system acquires $M = 100$ repetitions at any (x_0, y_0) position and moves to the next $(x_0, y_0 + \Delta y)$ position every exact 6 ms ($M \cdot T_s + 2$ ms). Therefore, we ensure phase synchronization by propagating precisely 12 cycles of the 2 kHz harmonic signal every 6 ms. For the Lamb wave case, one cycle of 1 kHz is burst at every (x, y) position every time a trigger signal is received from the PC controlling the OCT acquisition.

The authors should address the effects of elastic anisotropy.

As reported in other research papers, the cornea tissue is anisotropic and shows differentiated wave speed along each *en face* layer. In this paper, we focus on the average speed per layer characterization of cornea along depth. Future work will concentrate on addressing the anisotropic characterization of cornea by increasing the excitation frequency (to create elastographic resolution). Further comments on this have been added in the sixth paragraph of Section 3, along with references for corneal anisotropy.

I would like to see additional discussion on passive elastography and the difference between reverberant shear waves and standing waves.

Comparison between passive elastography and our approach has been added at the end of the fifth paragraph in Section 3 as requested by the reviewer.

In the derivation in the supplemental materials, α is not defined. The authors also derive based on x , but then change the equations to z (the optical axis). I think that it would be much more straightforward if the derivations were performed with respect to z with a small preface that z is the optical axis with respect to the OCE system.

We added the definition of α in the second paragraph of Section 1.1 of the Supplementary Material. Regarding the theoretical derivation in that Section, the correlation axis is kept always along the z axis in order to produce closed form solutions and avoid mathematical complexities within the complex exponential term in the integral structure of Equation 7 and 11. Then, the optical axis (sensor axis) is aligned with the x -axis (for the perpendicular case), and z -axis (for the parallel case). Then we can accommodate the

final solutions to have the z-axis as the optical axis as is usually defined in PhS-OCT. This accommodation is possible because the material is assumed to be isotropic. Unfortunately, at this time, we cannot define a fixed optical z-axis and change the correlation axis and find closed-form solutions. Currently, we are extending the theoretical derivations for anisotropic materials and it will be material of a future publication.

The cornea was irrigated with 1X PBS. PBS is not isotonic with the cornea and will cause the cornea to swell. The authors should address this and any effects of cornea edema on their results.

This effect has been discussed at the end of the sixth paragraph in Section 3 as suggested by the reviewer.

In the Monte Carlo simulations, what do the standard deviations represent?

We do not have standard deviations in Section 2.1., regarding the free space Monte-Carlo simulation of reverberant fields. In the finite element simulations of Section 2.2, the standard deviations in Fig 2b are for samples across the transverse dimensions of each layer or depth as stated in the caption of Figure 2.

How did the authors determine which wavelengths are not supported by the tissue in the data processing?

In porcine cornea, we refer to publications [36,39,41,45,61] to provide a range of speeds between 0.2 m/s to 10 m/s supported by this tissue type under different excitation frequencies and IOP levels. In order to confirm these assumptions, we analyze the wave field after filtering for higher speeds (> 10 m/s) and we found noise for all IOP cases of porcine cornea experiments under 2 kHz. We have included this information in the second paragraph of Data Processing Scheme in Section 4.

What are the major limitations of this technique?

Major limitations of Rev3D-OCE have been discussed in the sixth paragraph in Section 3 as suggested by the reviewer.

There are some minor grammatical and typographical errors: “being the stroma is the thickest layer in the cornea”, “the needle is tubbed”

We corrected the typographical errors. thank you, as: “the stroma being the thickest layer in the cornea” in the second paragraph of Section 1, and “The needle is tubed” in Sample Preparation in Section 4.

REVIEWER #3

This manuscript demonstrates an optical coherence elastography (OCE) technique to measure the shear moduli of different layers in the cornea. This technique involves exciting reverberant acoustic waves in the cornea using several piezoelectrically vibrating tips around the limbus and imaging the speckle-like interference pattern of the waves using phase-sensitive optical coherence tomography. From a 2D spatial auto-correlation analysis of the acoustic displacement field at a specific frequency, the acoustic wavelength of each layer is calculated, from which

shear acoustic speed and modulus is determined. The general principle of this technique has been known. However, this manuscript reports its first application in conjunction with optical imaging. The novelty of this work is high.

In terms of significance, the ability to measure the elastic moduli of different layers in the cornea with high accuracy is nontrivial and has potential to be clinically useful for the assessment of corneal health and diseases, as well as refractive surgeries. Having said that, however, it is unclear to me whether the technique indeed allows for accurate, high-resolution elastography with practical advantages over alternative techniques such as ultrasound-excited shear wave or lamb wave OCE. The manuscript is lacking in careful quantitative validation of the technique, and there are reasons to be concerned that a part, if not the majority, of the depth-dependent features they observe may not be real and inaccurate.

We have provided validation of our theoretical framework using a Monte Carlo analysis and have compared experimental vs. finite element analysis results in porcine cornea. We do, however, acknowledge the reviewer's suggestion in providing more quantitative validation of the technique. We have conducted experiments in layered phantoms in order to characterize the axial elastography resolution of our technique in Section 2.3 (new Figure 3 added). In addition, we have increased the number of healthy porcine cornea ($N = 10$) samples during experiments for different IOP levels for further analysis in Section 2.5 (new Figure 6 added). Results strongly support our claims and, thanks to the reviewer's suggestion, enriched the analysis of the manuscript.

The key assumption is that acoustic waves are randomly mixed up and reach the so-called "diffuse" regime, where their speeds are close to the pure-shear speed. This important assumption needs to be verified experimentally. The authors may have different ideas how to validate. Perhaps they can start with a circular plate made of uniform material and then use corneal phantoms consisting of multiple layers of known elastic moduli. It should be noted that the effects of the acoustic attenuation and the boundary conditions also influence the quality of the reverberation pattern.

We have carried out experiments in a layered phantom to validate the presence of diffuse fields and to confirm the possibility of measuring single layer elasticity. Accuracy error $< 6.9\%$ has been found when Rev3D-OCE measurements in 3% and 5% gelatin concentration materials are compared to ground truth parameters (obtained through mechanical testing). Elastography resolution along the axial axis has been also characterized and reported in Section 2.3.

Some specific questions are:

1. The ideal diffuse regime requires acoustic waves in all directions to be excited with relatively equal magnitudes. To what extent was this state achieved? Show that the result is independent of the excitation geometry. Vary the phases and locations of the excitation points and show that the results (depth profiles) are consistent.

The derivation of reverberant fields assumes a uniform distribution (or probability density function) for the amplitude and phase of plane waves across all 4π steradians. This is less strict than requiring relatively equal magnitudes. In practice, our current 3D-printed pronged-ring is a single solid device connected to a piezo-electric transducer and the temporal excitation phase for each tip is the same. We are currently working on

the individual spatial and temporal control of each tip of the ring using small piezoelectric transducers, which will be reported in a subsequent publication. Experiments conducted in a layered phantom (Section 2.3) show good correlation between speed values and ground truth parameters using the current ring geometry. Future research will focus on studying different ring geometries and their impact on creating reverberant shear wave fields.

2. The technique analyzes layer by layer. Therefore, any noise and artifacts will show up as a depth profile. Quantitative validation with known samples and phantoms is needed.

We agree. We have added a layered phantom experiment in Section 2.3, where we compare our depth-dependent estimation against mechanical measurements (used in this paper as ground truth) for precision and accuracy characterization. Results (New Figure 3) show good agreement between mean Rev3D-OCE estimations and ground truth parameters. Variability of speed in the uniform regions is detected and attributed to noise, low number of wave cycles in the stiffer region, and the artifacts associated with the cross-correlation approach.

3. Likewise, any radial variation of a sample can affect the propagation of waves, and this can show up as depth dependence. To understand to what extent they contain artifacts, experiments with a layered phantom will be very helpful.

As described in the Data Processing Scheme subsection in Section 4, we take into account the curvature of the sample surface for the interpolation of motion planes in which we applied the auto-correlation and fitting to Equation 2. We conducted experiments in a layered phantom in order to analyze the variability of depth-dependent speed results along various independent measurements ($N = 5$) at different locations in the phantom. Results are in good agreement with ground truth parameters. Analysis of different surface curvatures and their impact on speed has not been explored in this research and would be material for a subsequent study.

4. In Fig. 2, it seems a constant speed was assumed in the FEM simulation, rather than simulating the generation of waves from the PZT contacts.

The finite element model in Figure 2 has 4 layers with differentiated elasticity distributed along depth, and the varying material parameter was accounted for. The type of simulation analysis is *steady-state dynamic direct* for a vibration of 2 kHz. This type of analysis runs a numerical solution of the wave field for the limiting case of a sinusoidal harmonic excitation and not a transient simulation such as in the Lamb wave case. If Figure 2 is zoomed properly, a gradient in the wavelength pattern can be seen along with depth. For better visualization, Figure 2 already has *en face* motion frames extracted at layers A, B, C, and D, depicting differentiated wavelength, and, therefore, speed.

5. Show raw data $B(x,y)$ measured with the cornea. The ROI in the sample was defined to be 8×8 mm. Analyze B for smaller ROI in x, y . How uniform is the result across the cornea?

We added the new Supplementary Figure 1 and analyzed the impact of the auto-correlation window size on the shear wave speed estimation process in corneal tissue. Results suggest that higher accuracy is achieved when the window size is increased (Supplementary Figure 1c). Examples of 2D autocorrelation plots, 2D shear wave speed

maps, and shear speed distribution are also shown for auto-correlation window sizes 2×2 , 3×3 , and 4×4 mm. We have added a discussion of this in the first paragraph of Section 2.4.

6. Does the spatial correlation along the x-axis versus y-axis produce the identical result?

In corneal tissue, auto-correlation profiles in the x -axis and y -axis are not identical as shown in Supplementary Figure 1, mainly attributed to the anisotropic mechanical properties of corneal layers. We have added a discussion of this in the first paragraph of Section 2.4.

7. Show measurement repeatability for the same cornea.

Figure 6a, showing depth-dependent shear wave speed profiles, was updated with measurements of variability for the same cornea under four IOP levels. The variability measurements (shaded bounds in every profile) were extracted from every single corneal layer as the standard error (SE) of the shear wave speed calculated along with that layer, which is not completely uniform due to anisotropic effects produced by the collagen distribution in the cornea. This explanation was also added in the second paragraph of Section 2.5.

8. Show data for several corneas with similar sizes and show inter-sample variability.

We have increased the number of cornea samples from 3 to 10 with similar sizes and shown inter-sample variability in Figure 6c. Herein, 6 zones were defined along with the corneal depth, and average shear wave speed values for all the samples are shown for every zone and IOP level. Analysis of these results and discussion has been added in the third paragraph of Section 2.5.

9. When each PZT excites Lamb wave (primarily lowest-order antisymmetric), how can 8 PZT end up generating random waves? Or, is the technique measuring interference of 8 Lamb waves?

Given the excitation frequency, we have demonstrated in numerical simulations and experiments in the cornea that diffuse fields are generated after the steady-state regime has been achieved. Such a regime implies the propagation of waves in different directions and not only following the corneal surface such as Lamb waves or other surface acoustic wave branches. In the fourth paragraph of Section 3, we have added a whole discussion of how parameters such as frequency (2 kHz), the number of tips in the ring (between 6 to 8), and the time to reach the diffuse field regime (~ 1 s) have been explored in simulations and selected for generating reverberant fields in porcine cornea.

10. The acoustic attenuation at 2 kHz is non-negligible. It may be that each wave makes no more than one reflection from the edges of the cornea and the extent of mode conversion may be small. To what extent the diffuse field assumption valid?

The attenuation of shear waves at 2 kHz in the cornea is non-negligible, but still produces sufficient displacement to be detected by an OCT system when analyzed in a ROI of 8×8 mm. In addition, the continuous sinusoidal excitation of the cornea stimulates the continuous propagation and reflection of shear waves in comparison with a single tone propagation (Lamb wave method) that rapidly loses its energy when propagating. Higher

frequencies will strongly attenuate the wave field, preventing its propagation and detection within the selected ROI. Alternatives to such situations will include reducing the ROI and moving the excitation tips closer to the ROI in order to reduce the propagation distance, and, therefore, the attenuation.

11. Please explain “Furthermore, if a transverse isotropic model of the material is assumed, G corresponds to out-of-plane shear modulus.”

The derivations of Equations 1, 2, and 3 were conducted assuming a uniform material. However, evidence demonstrates that the collagen distribution within the cornea fits best to the description of a transverse isotropic material model in which the shear modulus (in-plane shear modulus) is isotropic along any single corneal layer and different to the shear modulus along depth (out-of-plane shear modulus) according to terminology provided in [21]. In this case, since the OCT system is only measuring displacement along depth (out-of-plane displacement), our estimations are hypothesized to correspond to out-of-plane shear modulus. We are currently developing the theory to extend Equations 1, 2, and 3 to the anisotropic case, and this will be material for a next publication. We clarified this point at the end of Section 2.1 in the revised manuscript.

12. Is the method independent of sample thickness? Verify using phantoms of uniform material but different thicknesses.

According to our multi-layer phantom results, we found an axial elastography resolution of $55.5 \mu\text{m}$ (FWHM) as shown in Figure 3. The top layer has a thickness of $\sim 0.3 \text{ mm}$, while the bottom layer can be considered semi-infinite media. Therefore, the proposed technique is able to detect layers with thicknesses greater than $\sim 60 \mu\text{m}$. Unfortunately, we didn't conduct experiments with a wider range of phantom thicknesses. Extended research including this study and a more formal resolution characterization will be material for a subsequent publication.

13. The current method does not provide any transverse resolution. Reducing the size of ROI will decrease the quantitative accuracy. Keratoconus is usually a focal disease that may have mechanical inhomogeneity in both x , y , and z . This may be a limitation of this technique.

Indeed, transverse resolution is limited by the size of the autocorrelation window, set to $2 \times 2 \text{ mm}$ or higher in these studies. We are currently working on how to improve the technique in order to detect such gradients with a similar or improved resolution as along the axial axis. This limitation has been added in the sixth paragraph in Section 3 as suggested by the reviewer.

14. How smaller feature in z can be measured? At 2 kHz and $c_s = 1 \text{ m/s}$, the acoustic wavelength 2 mm , almost two orders of magnitude larger than the thickness of membrane in the cornea. Although the acoustic wave is not directly translated to the spatial resolution of elastography, a controlled experiment using a phantom containing a thin layer with various thickness can demonstrate spatial resolution in z .

As suggested by the reviewer, we have included this study in a layered phantom in Section 2.3. Results show that our technique is able to reach a $55.5 \mu\text{m} \pm 17.3 \mu\text{m}$ axial

elastography resolution, which validates detection of corneal layers. Also, an extended study with phantom layers of different sizes will be material for a later publication.

In summary, we have addressed each of the reviewers' suggestions, and these have been very helpful. Thank you for the editorial work.

REVIEWERS' COMMENTS:

Reviewer #1 (Remarks to the Author):

The authors satisfactory addressed all my previous suggestions and comments.

Reviewer #3 (Remarks to the Author):

The revised manuscript addressed all my questions and suggestions that were raised. Great work. I recommend Accept as is.